# Discrete GCBF Proximal Policy Optimization for Multi-agent Safe Optimal Control

**Songyuan Zhang**[†]   **Oswin So**[†]   **Mitchell Black**[*]   **Chuchu Fan**[†]
[†]Department of Aeronautics and Astronautics, MIT    [*]MIT Lincoln Laboratory
[†]{szhang21,oswinso,chuchu}@mit.edu   [*]mitchell.black@ll.mit.edu

## Abstract

Control policies that can achieve high task performance and satisfy safety constraints are desirable for any system, including multi-agent systems (MAS). One promising technique for ensuring the safety of MAS is distributed control barrier functions (CBF). However, it is difficult to design distributed CBF-based policies for MAS that can tackle unknown discrete-time dynamics, partial observability, changing neighborhoods, and input constraints, especially when a distributed high-performance nominal policy that can achieve the task is unavailable. To tackle these challenges, we propose **DGPPO**, a new framework that *simultaneously* learns both a *discrete* graph CBF which handles neighborhood changes and input constraints, and a distributed high-performance safe policy for MAS with unknown discrete-time dynamics. We empirically validate our claims on a suite of multi-agent tasks spanning three different simulation engines. The results suggest that, compared with existing methods, our DGPPO framework obtains policies that achieve high task performance (matching baselines that ignore the safety constraints), and high safety rates (matching the most conservative baselines), with a *constant* set of hyperparameters across all environments.[1][2]

## 1 Introduction

Multi-agent systems (MAS) have gained significant attention in recent years due to their potential applications in various domains, such as warehouse robotics (Kattepur et al., 2018), autonomous vehicles (Shalev-Shwartz et al., 2016), traffic routing (Wu et al., 2020), and power systems (Biagioni et al., 2022). However, a big challenge for MAS is designing distributed control policies that can achieve high task performance while ensuring safety, especially when the two are conflicting. In the single-agent continuous-time case, control barrier functions (CBF) are an effective tool to resolve the conflict via the solution of a safety filter quadratic program (QP) (Xu et al., 2015; Ames et al., 2017), minimally modifying a given performance-oriented nominal policy to be safe. While distributed CBFs have been proposed for multi-agent (Wang et al., 2017) and partially observable cases (Zhang et al., 2025), they have a limitation of requiring *known* continuous-time dynamics and a nominal policy that can achieve high task performance (albeit not necessarily safely).

While the assumptions above are reasonable for many applications, they do not apply when the dynamics are *unknown* and a performance-oriented nominal policy is unavailable. The challenge of requiring a nominal policy has been addressed by approaches that combine CBFs with reinforcement learning (RL) (Cheng et al., 2019; Emam et al., 2022), where the nominal policy is learned via an unconstrained RL algorithm to maximize task performance while the CBF is used as a safety filter to ensure safety. However, these works have only been applied to the single-agent case, and require known control-affine dynamics to ensure the resulting safety filter QP is computationally tractable, which is too strict for most systems, e.g., with contact dynamics, especially in discrete time.

A third challenge is that CBF-based methods require a CBF to be known. This can be challenging in the case of input constraints since not every function satisfies the CBF conditions (Chen et al., 2021). Constructing a CBF is even more challenging in the case of MAS with changing neighborhoods and limited sensing (Zhang et al., 2025). Zhang et al. (2025) proposed a learning framework for

---

[1]DISTRIBUTION STATEMENT A. Approved for public release. Distribution is unlimited.
[2]Project website: `https://mit-realm.github.io/dgppo/`

constructing a graph CBF (GCBF) for MAS that guarantees safety while satisfying input constraints. However, they assume known continuous-time dynamics and require an existing nominal policy.

In this work, we address these challenges by proposing a novel framework that simultaneously learns a discrete graph CBF and a high-performance safe policy for a MAS under unknown discrete-time dynamics, changing neighborhoods, and input constraints. We summarize our contributions below.

- We propose a method of learning discrete CBFs (DCBF) for unknown discrete-time dynamics and with input constraints.
- We propose the discrete GCBF (DGCBF), a discrete-time extension of the GCBF, for ensuring safety under varying neighborhoods in the limited sensing setting and extend the DCBF learning above to the case of DGCBF.
- We propose **D**iscrete **G**raph CBF **P**roximal **P**olicy **O**ptimization (**DGPPO**), a framework combining RL and DGCBF for solving discrete-time multi-agent safe optimal control problems for MAS with unknown dynamics and limiting sensing without a known performant nominal policy.
- Through extensive simulations, we demonstrate that DGPPO outperforms existing methods and is not sensitive to hyperparameters. Specifically, compared to existing methods that require different choices of hyperparameters per environment, DGPPO achieves the lowest cost compared to baselines with near $100\%$ safety rate using a single set of hyperparameters.

## 2 RELATED WORK

**Constructing Decentralized CBFs.** The challenge of applying CBFs for MAS has been explored via the construction of distributed CBFs (Borrmann et al., 2015; Glotfelter et al., 2017; Wang et al., 2017; Lindemann & Dimarogonas, 2019; Black & Panagou, 2023), which only take local observations as input. This simplifies the big centralized QP problem into small QP problems to be solved per agent. However, the construction is either limited to the case of unbounded control (Lindemann & Dimarogonas, 2019) or only for a specific dynamics model (e.g., double integrator) (Borrmann et al., 2015; Glotfelter et al., 2017; Wang et al., 2017). Recent advances in *learning* CBFs using neural networks (Saveriano & Lee, 2019; Srinivasan et al., 2020; Lindemann et al., 2021; Peruffo et al., 2021; Dawson et al., 2022; So et al., 2024; Knoedler et al., 2024) has resulted in works that investigate learning distributed CBFs (Qin et al., 2021; Zhang et al., 2023; 2025; Zinage et al., 2024). Nevertheless, these approaches assume *known* dynamics and are only applicable to *continuous-time* dynamics and hence cannot be applied to our problem setting. Moreover, it is assumed that a performant nominal policy is available, which we do not consider in this work.

**CBF in RL.** Originally inspired by the prospect of safety during training, recent works have integrated CBFs into the RL training process via the safety filter (Tearle et al., 2021; Hsu et al., 2023; Garg et al., 2024) for both single-agent (Cheng et al., 2019; Emam et al., 2022; Hailemichael et al., 2023) and multi-agent (Pereira et al., 2021; 2022) cases. Although both continuous-time (Emam et al., 2022; Hailemichael et al., 2023) and discrete-time (Cheng et al., 2019) dynamics have been considered, a major limitation is the requirement of affine (D)CBFs and control-affine dynamics up to a constant disturbance term to be learned. In contrast, the problem we tackle in this work does not make any such assumptions about the safety specifications or the structure of the dynamics.

**Safe Multi-agent RL.** The problem of constructing safe policies for MAS has also been studied in the RL community (Garg et al., 2024). Early works achieved safety via reward function design (Chen et al., 2017b;a; Long et al., 2018; Everett et al., 2018; Semnani et al., 2020). However, these approaches do not guarantee the satisfaction of the safety constraints even for the optimal policy (Massiani et al., 2023; Everett et al., 2018; Long et al., 2018). More recently, in the single-agent case, methods work with constraints in the form of the constrained Markov decision process (CMDP) problem and apply techniques from constrained optimization, including primal methods (Xu et al., 2021), primal-dual methods using Lagrange multipliers (Borkar, 2005; Tessler et al., 2019; He et al., 2023; Huang et al., 2024), and via trust-region-based approaches (Achiam et al., 2017; He et al., 2023). Of these, Lagrange-multiplier-based approaches are the most popular due to their simplicity, leading to multi-agent extensions (Gu et al., 2023; Liu et al., 2021b; Ding et al., 2023; Lu et al., 2021; Geng et al., 2023; Zhao et al., 2024). However, Lagrangian methods for CMDPs have been observed to have unstable training and convergence to poor policies when the constraint threshold is *zero* (Zanon & Gros, 2020; He et al., 2023; So & Fan, 2023; Ganai et al., 2024), which is the setting we target in this work.

## 3 PROBLEM SETTING AND PRELIMINARIES

### 3.1 MULTI-AGENT CONSTRAINED OPTIMAL CONTROL PROBLEM

Consider an $N$ agent MAS. We aim to solve distributed control policies that minimize a joint cost describing a desired task while staying safe. Let the state and control of agent $i$ at timestep $k$ be $x_i^k \in \mathcal{X}$ and $u_i^k \in \mathcal{U}$, where $x_i^k$ contains agent $i$'s position $p_i^k \in \mathcal{P}$. The joint state is defined as $\mathbf{x}^k :=$ $[x_1^k; \ldots; x_N^k; y^k] \in \boldsymbol{\mathcal{X}}$, where $y^k \in \mathcal{Y}$ is non-agent states (e.g., obstacles, goals). The joint action is defined with $\mathbf{u}^k := [u_1^k; \ldots, u_N^k] \in \boldsymbol{\mathcal{U}}$. We assume $\mathbf{x}$ follows the general discrete-time dynamics

$$\mathbf{x}^{k+1} = f(\mathbf{x}^k, \mathbf{u}^k), \tag{1}$$

where $f : \boldsymbol{\mathcal{X}} \times \boldsymbol{\mathcal{U}} \to \boldsymbol{\mathcal{X}}$ describes the joint dynamics and is *unknown*. We consider the setting where agents only observe objects within their sensing radius $R > 0$. Let $\mathcal{N}_i(\mathbf{x}) = \{j \mid \|p_j - p_i\| \leq R\}$ denote the *neighborhood* of agent $i$. At timestep $k$, agent $i$ only has access to local observation $o_i^k := O_i(\mathbf{x}^k) \in \mathcal{O}$ for observation function $O_i$:

$$O_i(\mathbf{x}^k) = \left( \{o_{ij}\}_{j \in \mathcal{N}_i}, \ o_i^y \right), \qquad o_{ij} := O^a(x_i, x_j), \quad o_i^y := O^y(x_i, y) \tag{2}$$

for inter-agent $O^a : \mathcal{X} \times \mathcal{X} \to \mathcal{O}_a$ and non-agent $O^y : \mathcal{X} \times \mathcal{Y} \to \mathcal{O}_y$ observation functions. Let the joint cost function describing the desired task be denoted as $l : \boldsymbol{\mathcal{X}} \times \boldsymbol{\mathcal{U}} \to \mathbb{R}$ [3]. Each agent has an avoid set defined as $\mathcal{A}_i := \{o_i \in \mathcal{O} \mid h_i^{(m)}(o_i) > 0, \forall m\}$ for the avoid functions $h_i^{(m)} : \mathcal{O} \to \mathbb{R}$, $m \in \{1, \ldots, M\}$, such that the agent is *unsafe* it it enters $\mathcal{A}_i$ any time in the trajectory. We want to learn distributed policies $\mu_i : \mathcal{O} \to \mathcal{U}$ that minimize the joint cost $l$ while avoiding the avoid set $\mathcal{A}_i$ at all times. Formally, denoting the joint policy by $\boldsymbol{\mu}(\mathbf{x}) = [\mu_1(o_1); \ldots; \mu_N(o_N)]$, we want to solve the following discrete-time distributed multi-agent safe optimal control problem (MASOCP):

$$\min_{\mu_1, \ldots, \mu_N} \quad \sum_{k=0}^{\infty} l(\mathbf{x}^k, \boldsymbol{\mu}(\mathbf{x}^k)) \tag{3a}$$

$$\text{s.t.} \quad \mathbf{x}^{k+1} = f(\mathbf{x}^k, \boldsymbol{\mu}(\mathbf{x}^k)), \qquad\qquad \forall i \in \{1, \ldots, N\}, \, k \geq 0, \tag{3b}$$

$$h_i^{(m)}(o_i^k) \leq 0, \quad o_i^k = O_i(\mathbf{x}^k), \quad \forall i \in \{1, \ldots, N\}, \forall m \in \{1, \ldots, M\}, \, k \geq 0. \tag{3c}$$

The main challenge in solving (3) is satisfying the safety constraints (3c), especially in the multi-agent case with changing neighborhood, under unknown discrete-time dynamics with input constraints. We propose to tackle this challenge using the framework of DCBFs, which we review next.

### 3.2 DISCRETE CBF

To tackle the safety constraint of Problem (3), we review the notion of DCBF. Considering the discrete-time dynamics, we take the following definition of a DCBF from Ahmadi et al. (2019):

**Definition 1.** *A function $B : \boldsymbol{\mathcal{X}} \to \mathbb{R}$ is a discrete CBF (DCBF) for (1) if there exists an extended class-$\kappa$ function $\alpha$ satisfying $\alpha(-r) > -r$ for all $r > 0$ such that $B$ satisfies the following property:*

$$B(\mathbf{x}) \leq 0 \implies \inf_{\mathbf{u} \in \boldsymbol{\mathcal{U}}} B(f(\mathbf{x}, \mathbf{u})) - B(\mathbf{x}) + \alpha(B(\mathbf{x})) \leq 0. \tag{4}$$

As shown in Ahmadi et al. (2019), the following theorem holds.

**Theorem 1.** *The set $\mathcal{C} := \{\mathbf{x} \mid B(\mathbf{x}) \leq 0\}$ is control invariant under any policy $\boldsymbol{\mu}$ that satisfies*

$$B(f(\mathbf{x}, \boldsymbol{\mu}(\mathbf{x}))) - B(\mathbf{x}) + \alpha(B(\mathbf{x})) \leq 0, \quad \forall \mathbf{x} \in \mathcal{C} \tag{5}$$

Thus, if $\mathcal{C} \cap \mathcal{A} = \emptyset$ for the avoid set $\mathcal{A}$, then the $\boldsymbol{\mu}$ from Theorem 1 renders the system safe.

**Safe and Performant Policies via Safety Filtering.** Given DCBFs $B^{(m)}$ for $m = 1, \ldots, M$, we can construct a safe and performant policy using the safety filter framework. Assuming a nominal policy $\boldsymbol{\mu}_{\text{nom}}$ that is performant, e.g., minimizes the cost $l$, but not necessarily safe, we can obtain a safe *and* performant policy by solving the following nonlinear optimization problem:

$$\min_{\mathbf{u} \in \boldsymbol{\mathcal{U}}} \quad \|\mathbf{u} - \boldsymbol{\mu}_{\text{nom}}(\mathbf{x})\|^2, \tag{6a}$$

$$\text{s.t.} \quad C^{(m)}(\mathbf{x}, \mathbf{u}) \leq 0, \quad C^{(m)}(\mathbf{x}, \mathbf{u}) := B^{(m)}(f(\mathbf{x}, \mathbf{u})) - B^{(m)}(\mathbf{x}) + \alpha(B^{(m)}(\mathbf{x})), \quad \forall m. \tag{6b}$$

---

[3] The cost function $l$ here is **not** the *cost* in CMDP. Rather, it corresponds to the *negative reward* in CMDP.

In the general case, even if the dynamics $f$ are known, (6) is potentially a nonlinear program that could be difficult to solve. In our problem setting, we assume $f$ to be *unknown*, which renders this approach infeasible. Moreover, applying the safety filter framework assumes access to a performant, *distributed* nominal policy $\boldsymbol{\mu}_{\mathrm{nom}}$, which we do not assume is available. Even if it were available, the safety filtering only minimizes the instantaneous deviation in control (6a) and can be *myopic*, potentially leading to liveness problems (Reis et al., 2020) and deadlocks (Jankovic et al., 2023).

Note that not every function satisfies (5) and is a DCBF. In continuous time with control-affine dynamics $f$, unbounded inputs are sufficient for any continuously differentiable function to be a CBF (Xiao & Belta, 2019) since the CBF condition will be *linear* in the controls. However, in discrete time, the DCBF condition (5) is potentially *nonlinear* in $\mathbf{u}$, making it difficult to construct a DCBF even with unbounded controls. While solutions have been proposed for the continuous-time case to construct valid CBFs under bounded inputs (Chen et al., 2021; So et al., 2024), the same is not true for DCBFs, potentially due to the requirement of solving a nonlinear program (6) even after such a DCBF has been found.

## 4 TACKLING CHALLENGES OF DCBFs FOR MASOCP WITH DGPPO

We now address four challenges of extending a DCBF-based approach to our problem setting and propose DGPPO, our framework for solving (3). **All proofs can be found in Appendix A.**

### 4.1 CONSTRAINT-VALUE FUNCTION IS DCBF

To construct a DCBF, we show that the constraint-value function of a policy *is* a DCBF, extending the insights of So et al. (2024) to discrete-time. This allows learning a DCBF with policy evaluation.

For an arbitrary function $\zeta : \boldsymbol{\mathcal{X}} \to \mathbb{R}$, let the avoid set be $\mathcal{A} \coloneqq \{\mathbf{x} \in \boldsymbol{\mathcal{X}} \mid \zeta(\mathbf{x}) > 0\}$. For a fixed deterministic policy $\boldsymbol{\mu}$, consider the constraint-value functions $V^{\zeta,\boldsymbol{\mu}}$, defined as

$$V^{\zeta,\boldsymbol{\mu}}(\mathbf{x}) = \max_{k \geq 0} \zeta(\mathbf{x}^k), \qquad \text{s.t.} \quad \mathbf{x}^0 = \mathbf{x}, \quad \mathbf{x}^{k+1} = f(\mathbf{x}^k, \boldsymbol{\mu}(\mathbf{x}^k)). \tag{7}$$

Then, $V^{\zeta,\boldsymbol{\mu}}$ is a DCBF, which we show in the following theorem.

**Theorem 2** (Discrete Policy CBF). *For a given $\boldsymbol{\mu}$, the constraint-value function $V^{h,\boldsymbol{\mu}}$ is a DCBF for any extended class-$\kappa$ function $\alpha$ satisfying $\alpha(-r) > -r$ for all $r > 0$. Moreover, given $h^{(m)}$, $\boldsymbol{\mu}$ satisfies $V^{h^{(m)},\boldsymbol{\mu}}(f(\mathbf{x}, \mathbf{u})) - V^{h^{(m)},\boldsymbol{\mu}}(\mathbf{x}) + \alpha(V^{h^{(m)},\boldsymbol{\mu}}(\mathbf{x})) \leq 0$ for all $m \in \{1, \ldots, M\}$.*

Theorem 2 enables the construction of a DCBF by choosing any policy $\boldsymbol{\mu}$ and evaluating its constraint-value function. Consequently, we can construct a DCBF by learning the value function.

### 4.2 REMOVING THE NOMINAL POLICY WITH EXPLICIT COST OPTIMIZATION

We next address the challenge of requiring a performant nominal policy in the safety filter (6) by instead directly learning a policy using RL that minimizes the joint cost function (3a). This has been done previously in the single-agent setting via the framework of shielding for RL, where an unconstrained policy $\boldsymbol{\mu}_\theta$ with parameters $\theta$ is learned using existing unconstrained RL techniques, and the (D)CBF safety filter is incorporated into the environment dynamics (Cheng et al., 2019; Emam et al., 2022; Hailemichael et al., 2023). Formally, the following problem is considered:

$$\min_{\boldsymbol{\mu}_\theta} \quad \sum_{k=0}^{\infty} l(\mathbf{x}^k, \mathbf{u}^k), \qquad \text{s.t.} \quad \mathbf{u}^k = \mathrm{SafetyFilter}(\boldsymbol{\mu}_\theta(\mathbf{x}^k)), \tag{8}$$

where $\mathrm{SafetyFilter}$ computes the minimizer of (6). However, solving (6) in the discrete case is a nonlinear program that is difficult unless $B^{(m)}$ is linear and the dynamics are control-affine, an assumption that we, unlike previous works (Cheng et al., 2019; Emam et al., 2022; Hailemichael et al., 2023), do not impose. To work around this, we constrain the learned policy to satisfy the DCBF conditions instead of using the safety filter framework.

$$\min_\theta \sum_{k=0}^{\infty} l(\mathbf{x}^k, \boldsymbol{\mu}_\theta(\mathbf{x}^k)), \quad \text{(9a)} \quad \text{s.t.} \quad C^{(m)}(\mathbf{x}^k, \boldsymbol{\mu}_\theta(\mathbf{x}^k)) \leq 0, \quad \forall m = \{1, \ldots, M\}, \quad k \geq 0, \quad \text{(9b)}$$

where $C^{(m)}$ is defined in (6b). This removes the need for a nominal policy and for solving the nonlinear safety filter program (6).

### 4.3 Constrained Policy Optimization using DCBF Under Unknown Dynamics

We now tackle the challenge of performing constrained policy optimization using DCBFs in (9). We choose to use a purely primal method inspired by constraint-rectified policy optimization (CRPO) (Xu et al., 2021) due to its simplicity (it does not have parameters related to the dual problem but simply chooses to take different gradient steps based on the current constraint satisfaction). Specifically, if the constraint (9b) is satisfied, our algorithm takes one gradient step to minimize the objective (9a). Otherwise, if the constraint is violated, our algorithm takes one gradient step to minimize the DCBF constraint violation $C$.

This procedure still cannot be implemented as-is since the gradient of $C$ cannot be computed directly without knowledge of the dynamics $f$. To this end, we propose to use score function gradients (Williams, 1992) to compute gradients of (9b) without knowing the dynamics $f$. Since this requires a *stochastic* policy, we modify (9) accordingly. For clarity, let $\pi_\theta(\cdot|\mathbf{x})$ denote the probability density function of the stochastic policy with parameters $\theta$ conditioned on state $\mathbf{x}$. It can be tempting to consider the following stochastic version of the problem, which resembles the CMDP setting.

$$\min_\theta \quad \mathbb{E}_{\mathbf{x}\sim\rho_0,\mathbf{u}\sim\pi_\theta(\cdot|\mathbf{x})}\big[Q^{\pi_\theta}(\mathbf{x},\mathbf{u})\big], \qquad \text{s.t.} \quad \mathbb{E}_{\mathbf{x}\sim\rho^{\pi_\theta},\mathbf{u}\sim\pi_\theta(\cdot|\mathbf{x})}\Big[C^{(m)}(\mathbf{x},\mathbf{u})\Big] \leq 0, \qquad (10)$$

where $Q^{\pi_\theta}$ is the Q-function and $\rho_0$, $\rho^{\pi_\theta}$ are the initial and stationary state distributions. Here, we constrain the *expectation* of the DCBF constraint. However, this formulation is *not* sufficient for safety, as the expectation does not guarantee satisfaction for all states as in (6), leading to an unsafe policy. To tackle this, we modify the constraint, leading to the following problem.

$$\min_\theta \quad \mathbb{E}_{\mathbf{x}\sim\rho_0,\mathbf{u}\sim\pi_\theta(\cdot|\mathbf{x})}\big[Q^{\pi_\theta}(\mathbf{x},\mathbf{u})\big], \tag{11a}$$

$$\text{s.t.} \quad \mathbb{E}_{\mathbf{x}\sim\rho^{\pi_\theta}} \underbrace{\mathbb{E}_{\mathbf{u}\sim\pi_\theta(\cdot|\mathbf{x})}\Big[\max\big\{0,C^{(m)}(\mathbf{x},\mathbf{u})\big\}\Big]}_{:=\tilde{C}_\theta^{(m)}(\mathbf{x})} \leq 0, \quad \forall m. \tag{11b}$$

Here, the satisfaction of (11b) guarantees that the DCBF constraint is satisfied almost surely (Appendix A.2). Applying gradient-manipulation style primal optimization as in CRPO (Xu et al., 2021) gives us the following expression for the gradient $\nabla_\theta L$ of the policy loss $L$.

$$\nabla_\theta L(\theta) = \begin{cases} \nabla_\theta \mathbb{E}_{\mathbf{x}\sim\rho_0,\mathbf{u}\sim\pi_\theta(\cdot|\mathbf{x})}\big[Q^{\pi_\theta}(\mathbf{x},\mathbf{u})\big], & \mathbb{E}_{\mathbf{x}\sim\rho^{\pi_\theta}}\big[\tilde{C}_\theta^{(m)}(\mathbf{x})\big] \leq 0,\ \forall m, \\ \nu\,\mathbb{E}_{\mathbf{x}\sim\rho^{\pi_\theta}}\big[\nabla_\theta\tilde{C}_\theta^{(m)}(\mathbf{x})\big] \text{ for any } m \text{ that violates}, & \text{otherwise}, \end{cases}$$
$$(12)$$

where $\nu > 0$ is a hyperparameter scaling the size of constraint minimization steps. The gradient of $\tilde{C}_\theta^{(m)}$ can be computed using score function gradients (see Appendix A.4, discussion on $\rho^{\pi_\theta}$ in Appendix B.2).

**Improving sample efficiency with Gradient Projection.** One drawback of (12) is that this scheme is sample-inefficient in the sense that if only a single state $\bar{\mathbf{x}}$ violates the DCBF constraint, then the gradient information of the total cost from all other safe states are thrown away. We propose to use this thrown-away gradient information by *projecting* the cost gradient to be *orthogonal* to the gradient direction of the violating constraints, similar to techniques from multi-objective optimization (Yu et al., 2020; Liu et al., 2021a). However, this requires computing the gradient $M + 1$ times, which is expensive when $M$ is large. Instead, we propose to use the following informal theorem:

**Informal Theorem 3** (Approximate Gradient Projection for Decoupled Policy Parameters)**.** *Let* $\sigma^{(m)} := \nabla_\theta\mathbb{E}_{\mathbf{x}\sim\rho}[\tilde{C}_\theta^{(m)}(\mathbf{x})]$ *denote the gradient of the $m$-th DCBF constraint violation for any state distribution $\rho$. Under suitable assumptions on the policy parametrization $\pi_\theta$, modifying the gradient of the objective (11a) from* $g_{orig} := \mathbb{E}_{\mathbf{x}\sim\rho^{\pi_\theta}}\mathbb{E}_{\mathbf{u}\sim\pi_\theta(\cdot|\mathbf{x})}\big[\nabla_\theta\log\pi(\mathbf{x},\mathbf{u})\,Q^{\pi_\theta}(\mathbf{x},\mathbf{u})\big]$ *to*

$$g := \mathbb{E}_{\mathbf{x}\sim\rho^{\pi_\theta}}\mathbb{E}_{\mathbf{u}\sim\pi_\theta(\cdot|\mathbf{x})}\Big[\nabla_\theta\log\pi(\mathbf{x},\mathbf{u})\,\mathbb{1}_{\{\max_m\tilde{C}_\theta^{(m)}(\mathbf{x})\leq 0\}}\,Q^{\pi_\theta}(\mathbf{x},\mathbf{u})\Big], \tag{13}$$

*by multiplying the $Q^{\pi_\theta}$ with an indicator function gives an approximate projection $g$ of $g_{orig}$ such that $g \cdot \sigma^{(m)} = 0\ \forall m$, i.e., it lies in the orthogonal complement of the constraint gradients $\sigma^{(m)}$.*

We state this more formally in Theorem A2. By Informal Theorem 3, we can treat $g$ as an approximate projection of the gradient of the objective (11a) so that it does not interfere with

the gradients from the safety constraints. Combining $\sigma^{(m)}$ (with $\rho = \rho^{\pi_\theta}$) and $g$ from Informal Theorem 3 gives the following gradient $\nabla_\theta L$ of the policy loss $L$, where $\psi$ denotes the *stop gradient* operation (see Appendix B for a detailed derivation and discussion about important details).

$$L(\theta) = \mathbb{E}_{\mathbf{x} \sim \psi(\rho^{\pi_\theta})} \mathbb{E}_{\mathbf{u} \sim \psi(\pi_\theta(\cdot|\mathbf{x}))} \big[ \log \pi_\theta(\mathbf{x}, \mathbf{u}) \psi\big(\tilde{Q}(\mathbf{x}, \mathbf{u}, \theta)\big) \big], \tag{14}$$

$$\tilde{Q}(\mathbf{x}, \mathbf{u}, \theta) := \mathbb{1}_{\{\max_m \tilde{C}_\theta^{(m)}(\mathbf{x}) \leq 0\}} \psi(Q^{\pi_\theta}(\mathbf{x}, \mathbf{u})) + \nu \max_m \tilde{C}^{(m)}(\mathbf{x}, \mathbf{u}). \tag{15}$$

Although we cannot guarantee that the conditions of Informal Theorem 3 hold, we find that adding this gradient projection improves performance. We provide empirical comparisons between this gradient projection method (14) and methods without projection (10), (11) in Appendix C.6.1.

**Scheduling the weight $\nu$.** Empirically, different values of $\nu$ result in a tradeoff between safety and cost minimization (potentially due to not satisfying Informal Theorem 3). To remedy this, we schedule $\nu$ by taking $\nu = 1$ initially to encourage exploration at the beginning, then doubling it after $50\%$ and $75\%$ of the total update steps to emphasize safety. We investigate this further in Section 5.3.

### 4.4 EXTENDING TO THE MULTI-AGENT CASE WITH DGCBF

Having proposed three solutions to tackle the difficulty of using CBFs in RL for unknown discrete-time dynamics, we now tackle the final challenge of changing neighborhoods due to the limited sensing radius of each agent. For this, we draw inspiration from GCBF (Zhang et al., 2025), which provides a theoretical framework for constructing distributed CBFs that can handle varying neighborhood sizes, albeit for the case of continuous-time dynamics.

To construct a distributed DCBF $B$, we need $B$ to be a function of each agent's local observation $o_i$ as opposed to the joint state $\mathbf{x}$. We make this concrete in the following definition.

**Definition 2** (Discrete GCBF). *A function $\tilde{B} : \mathcal{O} \to \mathbb{R}$ is a Discrete Graph CBF (DGCBF) if there exists a class-$\kappa$ function $\alpha$ with $\alpha(-r) > -r$ for all $r > 0$ and a control policy $\mu : \mathcal{O} \to \mathcal{U}$ satisfying*

$$\tilde{B}(o_j(\mathbf{x})) \leq 0, \ \forall j \implies \tilde{B}(o_i^+(\mathbf{x})) - \tilde{B}(o_i(\mathbf{x})) + \alpha(\tilde{B}(o_i(\mathbf{x}))) \leq 0, \quad \forall \mathbf{x} \in \mathcal{X}, \ \forall i, \tag{16}$$

*where $o_i^+(\mathbf{x}) = O_i(f(\mathbf{x}, \boldsymbol{\mu}(\mathbf{x})))$ and $\boldsymbol{\mu}$ denotes the resulting joint policy from each agent using $\mu$.*

Since a DCBF $B$ is defined for a MAS with a *fixed* size $N$, $B$ can not be used when $N$ changes. Given a DGCBF $\tilde{B}$, we can construct a DCBF $B$ as $B(\mathbf{x}) := \max_i \tilde{B}(o_i(\mathbf{x}))$ (Appendix A.6), thus $\tilde{B}$ guarantees safety. However, the *same* DGCBF $\tilde{B}$ can *also* be used to guarantee safety for *any* $N$, which we show in Appendix A.7.

**Remark 1** (Discontinuity due to neighborhood changes). *Unlike the continuous-time case (Zhang et al., 2025), which assumes that the GCBF is unaffected by agents at the sensing-radius boundary, DGCBF does not have this requirement. This is because the proof of safety in GCBF relies on continuity (w.r.t. time) during neighborhood changes. However, in the discrete-time case, the proof of safety only looks at finite differences and hence does not require the continuity of the DGCBF.*

Finding a function $\tilde{B}$ that satisfies (16) is nontrivial, especially in the case of neighborhood changes due to the limited sensing radius. Though we can learn a DGCBF using Section 4.1, it is unclear how the learned function satisfies (16) when the neighborhood of agents changes. One sufficient way for this to hold is to take advantage of the attention mechanism to place zero weights on the features corresponding to agents far enough away such that the value of $\tilde{B}$ is not affected too much by such agents.

We state this informally in the following theorem (see Appendix A.5 for the formal version).

**Informal Theorem 4** (Satisfying (16) during neighborhood changes). *Let $\xi_1$, $\xi_2$, and $\xi_3$ be functions that encode the input observations into some feature space. Suppose $\tilde{B}$ is of the form*

$$\tilde{B}(O_i(\mathbf{x})) = \xi_1 \left( \sum_{j \in \mathcal{N}_i} w(o_{ij}) \xi_2(o_{ij}), \ \xi_3(o_i^y) \right), \tag{17}$$

*where $w : \mathcal{O}_a \to \mathbb{R}$ is a weighting function that goes to 0 for observations $o_{ij}$ of agents that are far enough away. Then, under technical conditions on the dynamics, $\tilde{B}$ satisfies (16) and is a DGCBF.*

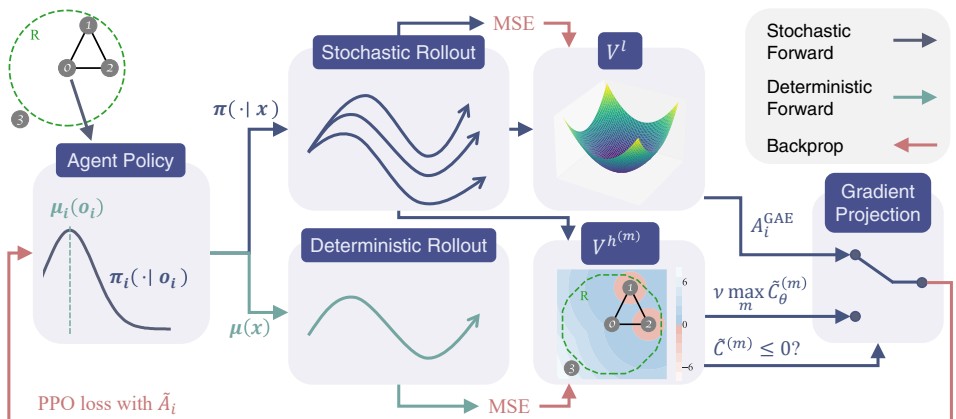

Figure 1: **DGPPO algorithm.** In addition to the normal MAPPO path (top) using **stochastic rollouts**, we introduce a second path (bottom) that uses **deterministic rollouts** to learn a DGCBF.

We encourage $\tilde{B}$ to satisfy (16) during neighborhood changes by parameterizing the value functions using graph neural networks (GNN) with graph attention (Veličković et al., 2017), which takes the form of (17) and hence is amenable to Informal Theorem 4. Similar to Zhang et al. (2025), the attention mechanism naturally learns to place zero weights on the features corresponding to neighboring agents that are far enough away, enabling the learned $\tilde{B}$ to satisfy (16) during neighborhood changes.

Finally, Theorem 2 can be extended to DGCBF, as shown in the following Corollary.

**Corollary 1** (Discrete Policy GCBF). *Suppose $V_i^{h^{(m)}, \boldsymbol{\mu}} : \mathcal{X} \to \mathbb{R}$ for agent $i$ can be expressed using only local observations $o_i$, i.e., there exists some $\tilde{V}^{h^{(m)}, \boldsymbol{\mu}} : \mathcal{O} \to \mathbb{R}$ such that*

$$\tilde{V}^{h^{(m)}, \boldsymbol{\mu}}(o_i^0) = V_i^{h^{(m)}, \boldsymbol{\mu}}(\mathbf{x}^0) := \max_{k \geq 0} h^{(m)}(o_i^k). \tag{18}$$

*Then, $\tilde{V}^{h^{(m)}}$ is a DGCBF.*

### 4.5 DGPPO: PUTTING EVERYTHING TOGETHER

Combining the proposed solutions from the previous subsections, we present **DGPPO**, a framework for solving the discrete-time multi-agent safe optimal control problem (3). DGPPO follows the basic structure of on-policy MARL algorithms such as MAPPO (Yu et al., 2022).

1. We perform a $T$-step stochastic rollout with the policy $\pi_\theta$. However, unlike MAPPO, we additionally perform a $T$-step *deterministic* rollout using a deterministic version of $\pi_\theta$ (by taking the mode), which we denote $\boldsymbol{\mu}$, to learn the DGCBF (per Theorem 2).
2. We update the value functions via regression on the corresponding targets computed using GAE (Schulman et al., 2015), where the targets for the cost-value function $V^l$ uses the stochastic rollout, and the targets for the constraint-value functions $V^{h^{(m)}, \boldsymbol{\mu}}$ use the deterministic rollout.
3. We update the policy $\pi_\theta$ by replacing the $Q$-function with its GAE (Schulman et al., 2015), then combining the CRPO-style *decoupled* policy loss (14) with the PPO clipped loss (Schulman et al., 2017) using the learned constraint-value functions $V^{h^{(m)}, \boldsymbol{\mu}}$ as the DGCBFs $\tilde{B}^{(m)}$. Specifically, we treat the expression within the expectation in (14) as a pseudo-advantage $\tilde{A}_i$ for agent $i$ and use a single-sample estimator $\hat{C}_{\theta,i}^{(m)}$ of $\tilde{C}_{\theta,i}^{(m)}$ in (14), giving us

$$\hat{C}_{\theta,i}^{(m)} := \max \left\{ 0, V^{h^{(m)}, \boldsymbol{\mu}}(o_i^+) - V^{h^{(m)}, \boldsymbol{\mu}}(o_i) + \alpha(V^{h^{(m)}, \boldsymbol{\mu}}(o_i)) \right\}, \tag{19}$$

$$\tilde{A}_i := A^{\text{GAE}} \mathbb{1}_{\{\max_m \hat{C}_{\theta,i}^{(m)} \leq 0\}} + \nu \max_m \hat{C}_{\theta,i}^{(m)} \mathbb{1}_{\{\max_m \hat{C}_{\theta,i}^{(m)} > 0\}} \tag{20}$$

where $A^{\text{GAE}}$ denotes the GAE (Schulman et al., 2015) for agent $i$. We then use $\tilde{A}_i$ in the PPO policy loss (Schulman et al., 2017) as done in MAPPO (Yu et al., 2022).

We summarize our DGPPO algorithm in Figure 1.

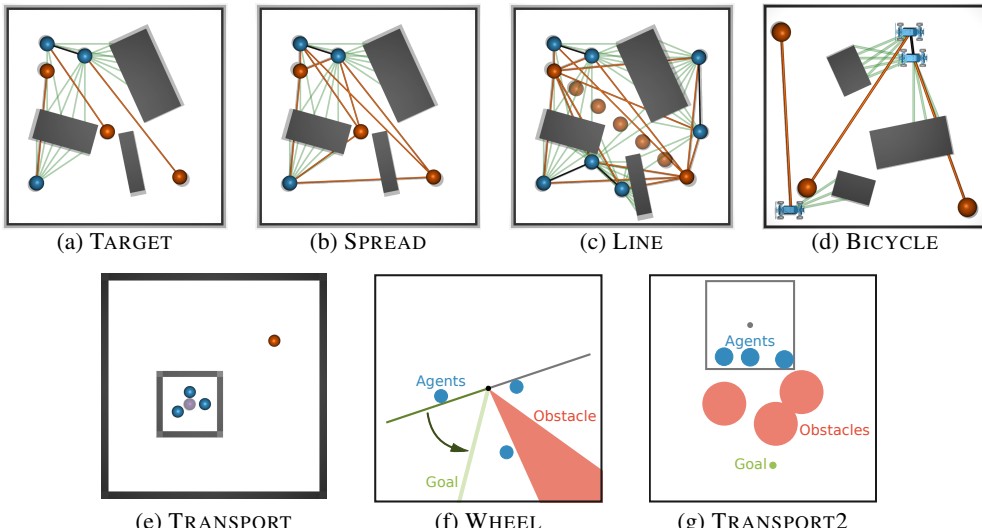

Figure 2: **Environments.** We test on (top) LiDAR, (bottom) MuJoCo, and VMAS environments.

## 5 EXPERIMENTS

In this section, we design experiments to answer the following research questions: **(Q1)** Does DGPPO learn a safe policy that also achieves low costs without hyperparameter tuning in different environments? **(Q2)** How stable is the training of DGPPO? **(Q3)** Can DGPPO maintain its performance with an increasing number of agents? **(Q4)** Is DGPPO sensitive to the hyperparameters?

To compare the methods, we look at the cost and safety rate. The **cost** is the trajectory cumulative cost $\sum_{k=0}^{T} l(\mathbf{x}^k, \mathbf{u}^k)$. The **safety rate** is the ratio of agents that are safe over the *entire* trajectory. Details on implementation, tasks, hyperparameters, and additional experiments are in Appendix C.

### 5.1 SETUP

**Environments.** We evaluate DGPPO in a wide range of environments including four LiDAR environments (TARGET, SPREAD, LINE, BICYCLE) where the agents use LiDAR to detect obstacles (Keyumarsi et al., 2023), one MuJoCo environment TRANSPORT (Todorov et al., 2012), and two VMAS environments (TRANSPORT2, WHEEL) (Bettini et al., 2022; 2024).

**Baselines.** We compare DGPPO against baseline methods that can solve MASOCP under unknown discrete-time dynamics, including the state-of-the-art MARL algorithm InforMARL (Nayak et al., 2023) and the safe MARL algorithm MAPPO-Lagrangian (Gu et al., 2021; 2023). For InforMARL, we add the constraint violations $\max\{0, \max_m h^{(m)}\}$ weighted by $\beta$ to the cost function for different $\beta$s (Penalty($\beta$)). We also try a weight-scheduling scheme where $\beta$ starts at $0.01$ and increases at $50\%$ and $75\%$ of the total steps (Schedule). For MAPPO-Lagrangian, we use a GNN backbone for fair comparison. We notice the official implementation (Gu et al., 2023) uses a tiny learning rate on the Lagrange multipliers ($10^{-7}$), so we consider two different initialization $\lambda_0$ (Lagr($\lambda_0$)). We also increase the learning rate of $\lambda$ to $0.1^4$ (Lagr(lr)). We run each method for the same number of update steps, chosen to be large enough such that all methods converge.[5]

### 5.2 MAIN RESULTS

For each environment, we run each algorithm with 3 different seeds and evaluate each run on 32 different initial conditions. We draw the following conclusions.

**(Q1): DGPPO has the best performance and is hyperparameter insensitive.** We first compare the converged policies of all algorithms (Figure 3). DGPPO is closest to the top left corner in all environments, indicating that it performs the best. For Penalty and Lagr, different choices of

---

[4]This is the smallest learning rate of $\lambda$ that does not make the algorithm ignore the safety constraints.

[5]We also test baselines with double environment steps similar to DGPPO for fairness (Appendix C.6.2).

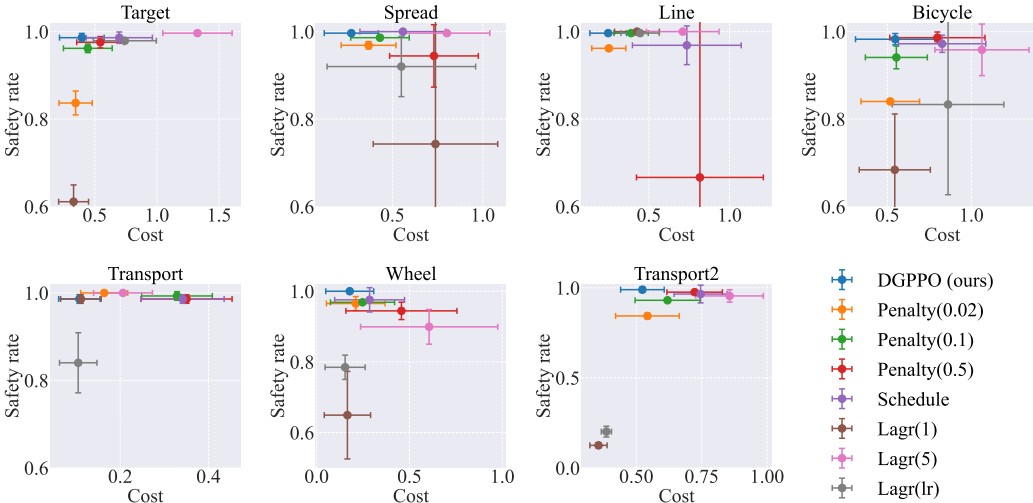

Figure 3: **Comparison on $N = 3$ agents.** ⊹ denotes the mean ± standard deviation. Methods closer to the top left yield lower costs and higher safety rates.

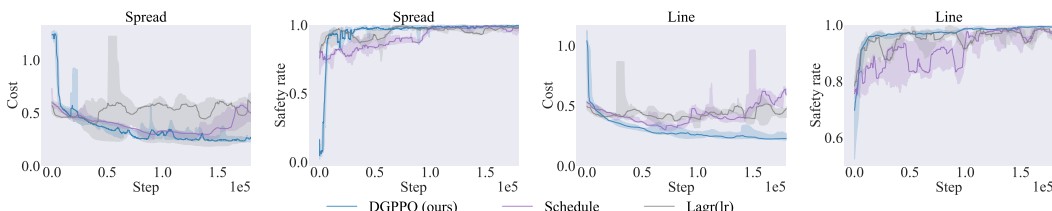

Figure 4: **Training stability.** DGPPO yields smoother training curves compared to the baselines.

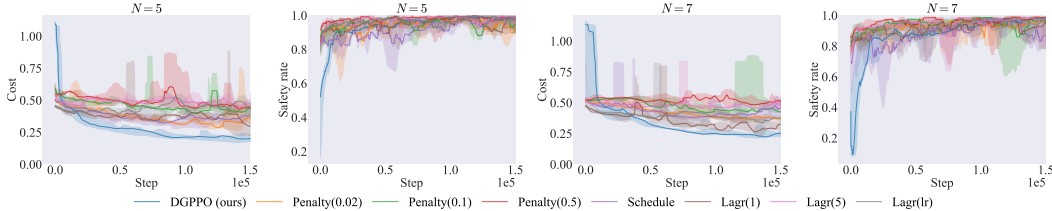

Figure 5: **Scaling to $N = 5, 7$.** Unlike other methods, DGPPO performs similarly with more agents.

hyperparameters result in either focusing only on safety or focusing only on performance. Even with a fixed hyperparameter, the performance of these two baselines also varies between environments. On the other hand, using the same set of hyperparameters in all environments, DGPPO consistently achieves the lowest cost among methods with a safety rate close to $100\%$.

**(Q2): Training of DGPPO is more stable.** Next, we compare the training stability of DGPPO with Schedule, due to having a weight scheduled, and Lagr(lr), due to having a non-negligible learning rate (Figure 4). Schedule experiences an increase in cost as $\beta$ increases throughout training. Lagr(lr) experiences high variance and many spikes in both cost and safety rate throughout training. This is similar to previous results obtained in the single-agent when the cost threshold is zero (So & Fan, 2023; He et al., 2023). DGPPO has a much smoother training curve than both. We provide training curves on other algorithms and environments in Appendix C.5.

**(Q3): DGPPO scales well with more agents.** Finally, we test the scalability of the methods on LINE by increasing the $N$ from 3 to 5 and 7 (Figure 5). The same trends from before still hold, with DGPPO achieving the best performance and high safety rates. We also see that DGPPO performs well even with more agents, but the baseline methods are more inconsistent (e.g., Schedule is mostly safe with $N = 5$ but not so for $N = 7$), possibly due to their hyperparameter sensitivity.

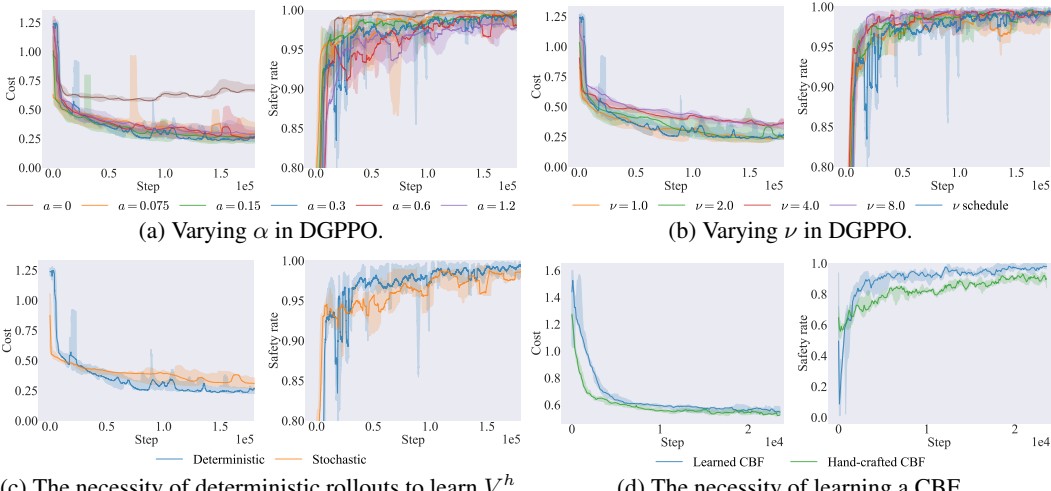

Figure 6: **Ablations.** We vary hyperparameters (top) and verify our design decisions (bottom).

## 5.3 ABLATION STUDIES

We now study hyperparameter sensitivity **(Q4)** by varying different hyperparameters in DGPPO.

**Class-$\kappa$ function $\alpha$.** For the class-$\kappa$ function in (16), we use a linear $\alpha(r) = ar$ with $a = 0.3$. We test the sensitivity of DGPPO to this by varying $a$ (Figure 6a) on SPREAD. We observe that $a = 0$ leads to conservative behavior with a high cost. $a = 1.2$ leads to an unsafe policy, which is to be expected since the $\alpha(-r) > -r$ condition is violated. For the other values that satisfy this condition, there is no significant difference in either cost or safety. We can thus choose any $a \in (0, 1)$.

**Weight $\nu$ on the gradient of $\tilde{C}$.** We introduced a schedule for $\nu$, which weights the constraint minimization step (Section 4.3). We test the sensitivity to $\nu$ on SPREAD with different static schedules (Figure 6b). The safety rate is lower with $\nu = 1$, while the convergence in cost for $\nu = 4, 8$ is slower. The proposed schedule leads to faster cost convergence and a high safety rate.

**Learning $V^h$ with a stochastic policy.** Theorem 2, used to learn the DGCBF $V^{h^{(m)},\boldsymbol{\mu}}$, requires *deterministic* rollouts. Consequently, DGPPO uses double the environment samples by performing both a stochastic and deterministic rollout. We verify whether this is necessary by seeing how the type of rollout (deterministic vs stochastic) used to learn the DGCBF affects performance (Figure 6c), which shows that using a stochastic rollout degrades both the cost and safety rate. Thus, the use of a deterministic rollout to learn $V^{h^{(m)},\boldsymbol{\mu}}$ is necessary despite the increased data use.

**Using a hand-crafted DGCBF.** One motivation for DGPPO is that it is difficult to construct a DGCBF with changing neighborhoods and input constraints. We test this by using $h^{(m)}$ *directly* as the DGCBF (instead of the learned $V^{h^{(m)},\boldsymbol{\mu}}$,), as is commonly done for CBFs, on TRANSPORT2 (Figure 6d). Using this hand-crafted "DGCBF" results in a decreased safety rate ($\sim 15\%$), validating the need to learn a DGCBF. If no DGCBF is used, it performs even worse (Appendix C.6.3).

## 6 CONCLUSION

We propose DGPPO to learn distributed safe policies for discrete-time MAS with unknown dynamics under a limited sensing range. We extend CBFs to this problem setting with DGCBFs, propose a construction using constraint-value functions, and apply CBFs to the case of unknown dynamics using score function gradients. Experimental results across three simulation engines suggest that DGPPO is robust to hyperparameters and performs well, achieving a safety rate matching conservative baselines while matching the performance of the performant but unsafe baselines.

**Limitations.** DGPPO uses both stochastic and deterministic rollouts, decreasing the sample efficiency. Moreover, safety under stochastic dynamics has not been considered. Finally, although safety is guaranteed when the DGCBF constraints are satisfied at *all* states, achieving this in practice using learning is hard. We leave these limitations to future work.

ACKNOWLEDGEMENT

This work was partly supported by the Under Secretary of Defense for Research and Engineering under Air Force Contract No. FA8702-15-D-0001. In addition, Zhang, So, and Fan are supported by the MIT-DSTA program. Any opinions, findings, conclusions, or recommendations expressed in this publication are those of the authors and don't necessarily reflect the views of the sponsors.

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

# A  PROOFS

## A.1  PROOF OF THEOREM 2

*Proof.* Dynamic programming on $V^{h,\mu}$ gives

$$V^{h,\mu}(x) = \max \left\{ h(x), \ V^{h,\mu}(f(x,\mu(x))) \right\}. \tag{21}$$

Let $V^{h,\mu}(x) \le 0$. This gives us two cases depending on which argument in the max is larger.

**Case 1:** $h(x) \le V^{h,\mu}(f(x,\mu(x)))$: Here, we have that $V^{h,\mu}(x) = V^{h,\mu}(f(x,\mu(x)))$, which implies

$$V^{h,\mu}(f(x,\mu(x))) - V^{h,\mu}(x) = 0 \le 0 \le -\alpha(V^{h,\mu}(x)). \tag{22}$$

**Case 2:** $h(x) > V^{h,\mu}(f(x,\mu(x)))$: Here, we have that $V^{h,\mu}(x) = h(x) > V^{h,\mu}(f(x,\mu(x)))$, which implies

$$V^{h,\mu}(f(x,\mu(x))) - V^{h,\mu}(x) < 0 \le 0 \le -\alpha(V^{h,\mu}(x)). \tag{23}$$

Thus, $V^{h,\mu}(f(x,\mu(x))) - V^{h,\mu}(x) + \alpha(V^{h,\mu}(x)) \le 0$ if $V^{h,\mu}(x) \le 0$, and $V^{h,\mu}$ is a DCBF. $\square$

## A.2  PROOF THAT EQUATION (11b) IMPLIES $C^{(m)}(\mathbf{x},\mathbf{u}) \le 0$ ALMOST SURELY

**Theorem A1.** *Suppose*

$$\mathbb{E}_{\mathbf{x}\sim\rho^{\pi_\theta}} \underbrace{\mathbb{E}_{\mathbf{u}\sim\pi_\theta(\cdot|\mathbf{x})} \left[ \max\left\{0, C^{(m)}(\mathbf{x},\mathbf{u})\right\} \right]}_{:=\tilde{C}_\theta^{(m)}(\mathbf{x})} \le 0. \tag{24}$$

*Then, $C^{(m)}(\mathbf{x},\mathbf{u}) \le 0$ almost surely.*

*Proof.* Note that (24) can only be satisfied when the expectation equals 0 since $\max\{0,\cdot\}$ is non-negative.

Assume for contradiction that $P(C^{(m)}(\mathbf{x},\mathbf{u}) \le 0) \le 1 - \epsilon$ for $\epsilon > 0$, i.e., $P(C^{(m)}(\mathbf{x},\mathbf{u}) > 0) \ge \epsilon$. Then,

$$0 = \mathbb{E}_{\mathbf{x}\sim\rho^{\pi_\theta}} \mathbb{E}_{\mathbf{u}\sim\pi_\theta(\cdot|\mathbf{x})} \left[ \max\left\{0, C^{(m)}(\mathbf{x},\mathbf{u})\right\} \right] \tag{25}$$

$$\ge P\left(C^{(m)}(\mathbf{x},\mathbf{u}) > 0\right) \mathbb{E}_{\mathbf{x}\sim\rho^{\pi_\theta},\mathbf{u}\sim\pi_\theta(\cdot|\mathbf{x})} \left[ \max\left\{0, C^{(m)}(\mathbf{x},\mathbf{u})\right\} \mid C^{(m)}(\mathbf{x},\mathbf{u}) > 0 \right] \tag{26}$$

$$= \epsilon \, \mathbb{E}_{\mathbf{x}\sim\rho^{\pi_\theta},\mathbf{u}\sim\pi_\theta(\cdot|\mathbf{x})} \left[ \max\left\{0, C^{(m)}(\mathbf{x},\mathbf{u})\right\} \mid C^{(m)}(\mathbf{x},\mathbf{u}) > 0 \right] \tag{27}$$

$$> 0. \tag{28}$$

which is a contradiction. Thus, $P(C^{(m)}(\mathbf{x},\mathbf{u}) \le 0) = 1$, and $C^{(m)}(\mathbf{x},\mathbf{u}) \le 0$ almost surely. $\square$

## A.3  FORMAL STATEMENT AND PROOF OF INFORMAL THEOREM 3

We first formally state Informal Theorem 3 below.

**Theorem A2** (Approximate Gradient Projection for Decoupled Policy Parameters). *Suppose that for all $\mathbf{x}_1 \ne \mathbf{x}_2$, the parameters $\theta$ of the stochastic policy $\pi_\theta$ are orthogonal, i.e., $(\nabla_\theta \pi_\theta(\mathbf{u}_1 \mid \mathbf{x}_1)) \cdot (\nabla_\theta \pi_\theta(\mathbf{u}_2 \mid \mathbf{x}_2)) = 0$ for all $\mathbf{u}_1, \mathbf{u}_2 \in \mathcal{U}$ (e.g., a finite state-space $\mathcal{X}$ with independent distribution at each state). Let $\sigma^{(m)} := \nabla_\theta \mathbb{E}_{\mathbf{x}\sim\rho}[\tilde{C}_\theta^{(m)}(\mathbf{x})]$ denote the gradient of the $m$-th DCBF constraint violation for any state distribution $\rho$. Then, the gradient of the objective (11a), modified with an extra indicator as follows:*

$$g := E_{\mathbf{x}\sim\rho^{\pi_\theta}} \left[ \mathbb{1}_{\{\max_m \tilde{C}_\theta^{(m)}(\mathbf{x})\le 0\}} \mathbb{E}_{\mathbf{u}\sim\pi_\theta(\cdot|\mathbf{x})} \left[ \nabla_\theta \log \pi(\mathbf{x},\mathbf{u}) \, Q^{\pi_\theta}(\mathbf{x},\mathbf{u}) \right] \right], \tag{29}$$

*satisfies $g \cdot \sigma^{(m)} = 0 \ \forall m$, i.e., it lies in the orthogonal complement of the constraint gradients $\sigma^{(m)}$.*

*Proof.* For $\mathbf{x} \in \mathcal{X}$, define the set $\Theta_{\mathbf{x}}$ as the column space of the gradient of the policy $\boldsymbol{\pi}$ at $\mathbf{x}$, i.e.,

$$\Theta_{\mathbf{x}} := \mathrm{span}\{\nabla_\theta \boldsymbol{\pi}_\theta(\mathbf{u} \mid \mathbf{x}) : \mathbf{u} \in \mathcal{U}\}. \tag{30}$$

By assumption, this implies that for $\mathbf{x}_1 \neq \mathbf{x}_2$,

$$\theta_1 \in \Theta_{\mathbf{x}_1}, \; \theta_2 \in \Theta_{\mathbf{x}_2} \implies \theta_1 \cdot \theta_2 = 0. \tag{31}$$

Now, note that

$$\sigma^{(m)} := \nabla_\theta \mathbb{E}_{\mathbf{x} \sim \rho}[\tilde{C}_\theta^{(m)}(\mathbf{x})] \tag{32}$$

$$= \nabla_\theta \mathbb{E}_{\mathbf{x} \sim \rho} \mathbb{E}_{\mathbf{u} \sim \boldsymbol{\pi}_\theta(\cdot|\mathbf{x})}[\max(0, C_\theta^{(m)}(\mathbf{x}, \mathbf{u}))] \tag{33}$$

$$= \mathbb{E}_{\mathbf{x} \sim \rho} \mathbb{E}_{\mathbf{u} \sim \boldsymbol{\pi}_\theta(\cdot|\mathbf{x})} \left[ \nabla_\theta \log \boldsymbol{\pi}_\theta(\mathbf{u} \mid \mathbf{x}) \max(0, C_\theta^{(m)}(\mathbf{x}, \mathbf{u})) \right] \tag{34}$$

$$= \mathbb{E}_{\mathbf{x} \sim \rho} \mathbb{E}_{\mathbf{u} \sim \boldsymbol{\pi}_\theta(\cdot|\mathbf{x})} \left[ \mathbb{1}_{\{C_\theta^{(m)}(\mathbf{x},\mathbf{u})>0\}} \nabla_\theta \log \boldsymbol{\pi}_\theta(\mathbf{u} \mid \mathbf{x}) C_\theta^{(m)}(\mathbf{x}, \mathbf{u}) \right] \tag{35}$$

$$\subseteq \bigcup_{\mathbf{x}:\mathcal{E}^{(m)}} \Theta_{\mathbf{x}}, \qquad \mathcal{E}^{(m)} := \left\{ \mathbf{x} \in \mathcal{X} : \operatorname*{ess\,sup}_{\mathbf{u} \in \mathcal{U}} C_\theta^{(m)}(\mathbf{x}, \mathbf{u}) > 0 \right\}, \tag{36}$$

where we have used the score function gradient estimator of $\tilde{C}_\theta$ (Appendix A.4). Similarly,

$$g := \mathbb{E}_{\mathbf{x} \sim \rho^{\boldsymbol{\pi}_\theta}, \mathbf{u} \sim \boldsymbol{\pi}_\theta(\cdot|\mathbf{x})} \left[ \nabla_\theta \log \boldsymbol{\pi}_\theta(\mathbf{u} \mid \mathbf{x}) \mathbb{1}_{\{\max_m \tilde{C}_\theta^{(m)}(\mathbf{x}) \leq 0\}} Q^{\boldsymbol{\pi}_\theta}(\mathbf{x}, \mathbf{u}) \right], \tag{37}$$

$$= \mathbb{E}_{\mathbf{x} \sim \rho^{\boldsymbol{\pi}_\theta}, \mathbf{u} \sim \boldsymbol{\pi}_\theta(\cdot|\mathbf{x})} \left[ \nabla_\theta \log \boldsymbol{\pi}_\theta(\mathbf{u} \mid \mathbf{x}) \mathbb{1}_{\{\max_m \operatorname{ess\,sup}_{\mathbf{u} \in \mathcal{U}} C_\theta^{(m)}(\mathbf{x}) \leq 0\}} Q^{\boldsymbol{\pi}_\theta}(\mathbf{x}, \mathbf{u}) \right], \tag{38}$$

$$\subseteq \bigcup_{\mathbf{x}:\mathcal{F}} \Theta_{\mathbf{x}}, \tag{39}$$

where

$$\mathcal{F} := \left\{ \mathbf{x} \in \mathcal{X} : \max_m \operatorname*{ess\,sup}_{\mathbf{u} \in \mathcal{U}} C_\theta^{(m)}(\mathbf{x}, \mathbf{u}) \leq 0 \right\}, \tag{40}$$

$$= \left\{ \mathbf{x} \in \mathcal{X} : \operatorname*{ess\,sup}_{\mathbf{u} \in \mathcal{U}} C_\theta^{(m)}(\mathbf{x}, \mathbf{u}) \leq 0, \; \forall m \right\}. \tag{41}$$

Since $\mathcal{E}^{(m)} \cap \mathcal{F} = \emptyset$ for all $m$, we have that $\sigma^{(m)} \cdot g = 0$ for all $m$. $\qquad\square$

## A.4 SCORE FUNCTION GRADIENT ESTIMATOR OF $\tilde{C}_\theta$

*Proof.* Using the log trick,

$$\nabla_\theta \mathbb{E}_{\mathbf{u} \sim \boldsymbol{\pi}_\theta(\mathbf{x})} \left[ C(\mathbf{x}, f(\mathbf{x}, \mathbf{u})) \right] = \nabla_\theta \int C(\mathbf{x}, f(\mathbf{x}, \mathbf{u})) \, \boldsymbol{\pi}_\theta(\mathbf{x}, \mathbf{u}) \, d\mathbf{u},$$

$$= \int C(\mathbf{x}, f(\mathbf{x}, \mathbf{u})) \left( \nabla_\theta \log \boldsymbol{\pi}_\theta(\mathbf{x}, \mathbf{u}) \right) \boldsymbol{\pi}_\theta(\mathbf{x}, \mathbf{u}) \, d\mathbf{u}, \tag{42}$$

$$= \mathbb{E}_{\mathbf{u} \sim \boldsymbol{\pi}_\theta(\mathbf{x})} \left[ \nabla \log \boldsymbol{\pi}_\theta(\mathbf{x}, \mathbf{u}) \, C(\mathbf{x}, f(\mathbf{x}, \mathbf{u})) \right].$$

$$\square$$

## A.5 FORMAL STATEMENT AND PROOF OF INFORMAL THEOREM 4

We first formally state Informal Theorem 4 below.

**Theorem A3** (Satisfying (16) during neighborhood changes). *Suppose that the maximum distance an agent can travel in a time step is $\bar{d}$. Let $\tilde{B}$ be of the following form.*

$$\tilde{B}(O_i(\mathbf{x})) = \xi_1 \left( \sum_{j \in \mathcal{N}_i} w(o_{ij}) \xi_2(o_{ij}), \; \xi_3(o_i^y) \right), \tag{43}$$

*where $\xi_1 : \mathbb{R}^{\rho_1} \times \mathbb{R}^{\rho_2} \to \mathbb{R}$, $\xi_2 : \mathcal{O}_a \to \mathbb{R}^{\rho_1}$, and $\xi_3 : \mathcal{O}_y \to \mathbb{R}^{\rho_2}$ encode the observations into some feature space, and $w : \mathcal{O}_a \to \mathbb{R}$ is a weighting function such that*

$$w(o_{ij}) = 0, \quad \text{for all } x_i, x_j \text{ such that } \|p_i - p_j\| \geq R - 2\bar{d}. \tag{44}$$

*If 1) $\tilde{B}$ satisfies (16) for all transitions where the neighborhood does not change, and 2) for any transition $\mathbf{x} \to \mathbf{x}^+$ with a neighborhood change $\mathcal{N}_i(\mathbf{x}) \neq \mathcal{N}_i(\mathbf{x}^+)$, there exists a transition $\bar{\mathbf{x}} \to \bar{\mathbf{x}}^+$ where all agents that either enter or leave the neighborhood (i.e., the complement of $\mathcal{N}_i(\mathbf{x}) \cap \mathcal{N}_i(\mathbf{x}^+)$) are moved outside the sensing radius, and all the remaining agents move identically in $\mathbf{x}$ and $\bar{\mathbf{x}}$, then $\tilde{B}$ is a DGCBF.*

*Proof.* Let $\mathbf{x}$ and $\mathbf{x}^+$ be consecutive states such that the neighborhood of agent $i$ changes, i.e., $\mathcal{N}_i(\mathbf{x}) \neq \mathcal{N}_i(\mathbf{x}^+)$. Let $E := \mathcal{N}_i(\mathbf{x}) \setminus \mathcal{N}_i(\mathbf{x}^+)$ and $F := \mathcal{N}_i(\mathbf{x}^+) \setminus \mathcal{N}_i(\mathbf{x})$ denote the set of agents that leave and enter the neighborhood of agent $i$, respectively. Since the maximum distance agents can travel in one timestep is $\bar{d}$, the distance between agents can change by at most $2\bar{d}$ in one timestep. Hence, all agents exiting the neighborhood are at least $R - 2\bar{d}$ away agent $i$, i.e.,

$$j \in E \implies \|p_i - p_j\| \geq R - 2\bar{d}. \tag{45}$$

Similarly, all agents entering the neighborhood are at least $R - 2\bar{d}$ away from agent $i$ at the $\mathbf{x}^+$, i.e.,

$$j \in F \implies \|p_i^+ - p_j^+\| \geq R - 2\bar{d}. \tag{46}$$

Hence, by (44), we have that $w(o_{ij}) = 0$ for all $j \in E$ and $w(o_{ij}^+) = 0$ for all $j \in F$.

Now, by asumption, there exists consecutive states $\bar{\mathbf{x}}$ and $\bar{\mathbf{x}}^+$ with the same neighborhood $\mathcal{N}_i(\bar{\mathbf{x}}) = \mathcal{N}_i(\bar{\mathbf{x}}^+) = \mathcal{N}_i(\mathbf{x}) \cap \mathcal{N}_i(\mathbf{x}^+)$, such that $\bar{x}_j = x_j$ and $\bar{x}_j^+ = x_j^+$ for $j \in \mathcal{N}_i(\bar{\mathbf{x}})$, and similarly for non-agent states $y = \bar{y}$, $y^+ = \bar{y}^+$. By definition of the observation function $O_i$, $\mathbf{x}$ and $\bar{\mathbf{x}}$ share the same observation except for the $o_{ij}$ for $j \in E$, and similarly for $\mathbf{x}^+$ and $\bar{\mathbf{x}}^+$ for $j \in F$. Since $w_{ij} = 0$ for all $j \in E \cup F$, the form of $\tilde{B}$ (43) implies that

$$\sum_{j \in \mathcal{N}_i} w(o_{ij})\xi_2(o_{ij}) = \sum_{j \in \mathcal{N}_i} w(\bar{o}_{ij})\xi_2(\bar{o}_{ij}) \tag{47}$$

$$\sum_{j \in \mathcal{N}_i} w(o_{ij}^+)\xi_2(o_{ij}) = \sum_{j \in \mathcal{N}_i} w(\bar{o}_{ij})\xi_2(\bar{o}_{ij}^+) \tag{48}$$

Hence, we must have that

$$\tilde{B}(O_i(\mathbf{x})) = \tilde{B}(O_i(\bar{\mathbf{x}})), \quad \tilde{B}(O_i(\mathbf{x}^+)) = \tilde{B}(O_i(\bar{\mathbf{x}}^+)). \tag{49}$$

By assumption, $\tilde{B}$ satisfies the DGCBF condition (16) for all transitions where the neighborhood does not change, which includes $\bar{\mathbf{x}} \to \bar{\mathbf{x}}^+$. Hence,

$$\tilde{B}(o_i^+) - \tilde{B}(o_i) + \alpha\left(\tilde{B}(o_i)\right) = \tilde{B}(\bar{o}_i^+) - \tilde{B}(\bar{o}_i) + \alpha\left(\tilde{B}(\bar{o}_i)\right) \tag{50}$$

$$\leq 0, \tag{51}$$

and $\tilde{B}$ also satisfies the DGCBF condition (16) for transitions where the neighborhood changes. Thus, $\tilde{B}$ is a DGCBF. $\qquad\square$

## A.6 PROOF THAT A DGCBF CAN BE USED TO CONSTRUCT A DCBF

**Theorem A4.** *For a $N$-agent MAS, define $B : \mathcal{X} \to \mathbb{R}$ as*

$$B(\mathbf{x}) := \max_i \tilde{B}(O_i(\mathbf{x})). \tag{52}$$

*Then, $B$ is a DCBF.*

*Proof.* Let $\mu : \mathcal{O} \to \mathbb{R}$ denote the per-agent control policy corresponding to the DGCBF $\tilde{B}$ in Definition 2, and let $\boldsymbol{\mu}$ denote the resulting joint control policy. Then, under $\boldsymbol{\mu}$, (16) implies that for any $i$,

$$\tilde{B}(o_i^+(\mathbf{x})) - \tilde{B}(o_i(\mathbf{x})) + \alpha(\tilde{B}(o_i(\mathbf{x}))) \leq 0, \quad \forall \mathbf{x} \in \mathcal{X}. \tag{53}$$

Now, for a given $\mathbf{x} \in \mathcal{X}$, let $\mathbf{x}^+ = f(\mathbf{x}, \boldsymbol{\mu}(\mathbf{x}))$ denote the next state following $\boldsymbol{\mu}$. Let $i_1$ and $i_2$ denote the index that maximizes (52) at $\mathbf{x}$ and $\mathbf{x}^+$ respectively, i.e.,

$$i_1 := \arg\min_i \tilde{B}(O_i(\mathbf{x})) \implies B(\mathbf{x}) = \tilde{B}(O_{i_1}(\mathbf{x})), \tag{54}$$

$$i_2 := \arg\min_i \tilde{B}(O_i(\mathbf{x}^+)) \implies B(\mathbf{x}^+) = \tilde{B}(O_{i_2}(\mathbf{x}^+)). \tag{55}$$

$$\tag{56}$$

Then, using (53) and the fact that $(1 - \alpha)$ is also an extended class-$\kappa$ function and thus is a monotonic function (Ahmadi et al., 2019):

$$0 \geq \tilde{B}(O_{i_2}(\mathbf{x}^+)) - \tilde{B}(O_{i_2}(\mathbf{x})) + \alpha(\tilde{B}(O_{i_2}(\mathbf{x}))), \tag{57}$$

$$= \tilde{B}(O_{i_2}(\mathbf{x}^+)) - (1 - \alpha) \circ \tilde{B}(O_{i_2}(\mathbf{x})), \tag{58}$$

$$\geq \tilde{B}(O_{i_2}(\mathbf{x}^+)) - (1 - \alpha) \circ \tilde{B}(O_{i_1}(\mathbf{x})), \tag{59}$$

$$= B(\mathbf{x}^+) - (1 - \alpha) \circ B(\mathbf{x}), \tag{60}$$

$$= B(\mathbf{x}^+) - B(\mathbf{x}) + \alpha(B(\mathbf{x})). \tag{61}$$

and (4) holds. Thus, $B$ is a DCBF. $\qquad\square$

Since $\tilde{B}$ enables the construction of a DCBF, this implies that $\mathcal{C}$, the zero sub-level set of $B$, i.e.,

$$\mathcal{C} \coloneqq \{\mathbf{x} \mid B(\mathbf{x}) \leq 0\} = \bigcap_i \{\mathbf{x} \mid \tilde{B}(o_i(\mathbf{x})) \leq 0\}, \tag{62}$$

is forward-invariant under $\boldsymbol{\mu}$ and hence control invariant.

### A.7 DGCBF HAS GENERALIZABLE SAFETY GUARANTEES

In this subsection, we prove that the same DGCBF $\tilde{B}$ from Definition 2 can guarantee the safety of a MAS with any number of agents $N$. We will do this by showing that there exists an $\bar{N}$ such that if $\tilde{B}$ satisfies the DGCBF conditions (16) for $\bar{N}$ agents, then the **same** DGCBF $\tilde{B}$ will also satisfy the DGCBF conditions (16).

Let the maximum distance an agent can travel in a time step is $\bar{d}$. Given a sensing radius $R$, define $\bar{N}$ as the maximum number of agents that can be located within a ball of radius $2R + 2\bar{d}$. For a $N$-agent MAS, let $\boldsymbol{\mathcal{X}}_N$ and $\boldsymbol{\mathcal{U}}_N$ denote the joint state and control space respectively, and let $f_N$ denote the corresponding dynamics function. For generalizability to an arbitrary number of agents, we make the additional assumption that the dynamics $f_N$ are decoupled for each agent, i.e.,

$$x_i^{k+1} = f_0(x^k, u^k), \tag{63}$$

for per-agent dynamics function $f_0$.

For a state $\mathbf{x} \in \boldsymbol{\mathcal{X}}_N$, let $\mathbf{x}_{i,\bar{N}} \in \boldsymbol{\mathcal{X}}_{\bar{N}}$ denote the restriction of $\mathbf{x}$ to that of agent $i$ and its $\bar{N} - 1$ closest neighbors. We then have the following theorem.

**Theorem A5.** *Let $\tilde{B}$ satisfy the DGCBF conditions (16) for $\bar{N}$ agents, i.e., there exists a class-$\kappa$ function $\alpha$ with $\alpha(-r) > -r$ for all $r > 0$ and a control policy $\mu : \mathcal{O} \to \mathcal{U}$ satisfying*

$$\tilde{B}(O_j(\tilde{\mathbf{x}})) \leq 0, \ \forall j \implies \tilde{B}(O_i^+(\tilde{\mathbf{x}})) - \tilde{B}(O_i(\tilde{\mathbf{x}})) + \alpha(\tilde{B}(O_i(\tilde{\mathbf{x}}))) \leq 0, \quad \forall \tilde{\mathbf{x}} \in \boldsymbol{\mathcal{X}}_{\bar{N}}, \ \forall i, \quad (64)$$

*Then, $\tilde{B}$ also satisfies the DGCBF conditions (16) for any $N > \bar{N}$ agents, i.e.,*

$$\tilde{B}(O_j(\mathbf{x})) \leq 0, \ \forall j \implies \tilde{B}(O_i^+(\mathbf{x})) - \tilde{B}(O_i(\mathbf{x})) + \alpha(\tilde{B}(O_i(\mathbf{x}))) \leq 0, \quad \forall \mathbf{x} \in \boldsymbol{\mathcal{X}}_N, \ \forall i, \quad (65)$$

We will prove Theorem A5 by showing that (65) holds because it can be reduced to the case of (64). For convenience, define $B$ as in Theorem A4 so that

$$\tilde{B}(o_j(\mathbf{x})) \leq 0, \ \forall j \quad \Longleftrightarrow \quad B(\mathbf{x}) \leq 0. \tag{66}$$

Before we prove Theorem A5, we first prove a few helpful lemmas.

**Lemma 1** (Restriction leaves observations of all agents within $R + 2\bar{d}$ invariant)**.** *For any $\mathbf{x} \in \boldsymbol{\mathcal{X}}_N$ such that $B(\mathbf{x}) \leq 0$, the restriction of $\mathbf{x}$ to that of agent $i$ and its $\bar{N} - 1$ closest neighbors leaves the observation of agent $i$ and all agent within $R + 2\bar{d}$ of $i$ invariant, i.e., for any $i$,*

$$\|p_j - p_i\| \leq R + 2\bar{d} \quad \Longrightarrow \quad O_j(\mathbf{x}) = O_j(\mathbf{x}_{i,\bar{N}}) \tag{67}$$

*Proof.* Since $B(\mathbf{x}) \leq 0$, by definition of $\bar{N}$ there can be no more than $\bar{N} - 1$ agents within radius of $2R + \bar{d}$ of agent $i$. Hence, $\mathbf{x}_{i,\bar{N}}$ will include all agents within a radius of $2R + \bar{d}$ of agent $i$.

By definition of the observation function $O_i$ (2),

$$O_i(\mathbf{x}) = \left( \{o_{ij}\}_{j \in \mathcal{N}_i(\mathbf{x})}, \ o_i^y \right). \tag{68}$$

$O_j(\mathbf{x})$ depends only on all agents $l$ that are at most $R$ away from $j$. Since $\mathbf{x}$ and $\mathbf{x}_{i,\bar{N}}$ agree on all agents up to $2R + 2\bar{d}$ away from $i$, this implies that for all $j$ that are $R + 2\bar{d}$ away from $i$,

$$\mathcal{N}_j(\mathbf{x}) = \mathcal{N}_j(\mathbf{x}_{i,\bar{N}}), \tag{69}$$

thus the observation is unchanged as well. $\qquad\square$

Next, we show that the observation of the *next* state for agent $i$ is also left unchanged after restriction.

**Lemma 2** (Restriction leaves the next observation for agent $i$ unchanged)**.** *For any* $\mathbf{x} \in \mathcal{X}_N$ *such that* $B(\mathbf{x}) \leq 0$, *let* $\mathbf{x}^+ = f_N(\mathbf{x}, \boldsymbol{\mu}(\mathbf{x}))$ *denote the next state under* $\boldsymbol{\mu}$ *for* $\mathbf{x}$, *and* $\tilde{\mathbf{x}}^+ = f_{\bar{N}}(\mathbf{x}_{i,\bar{N}}, \boldsymbol{\mu}(\mathbf{x}_{i,\bar{N}}))$ *the next state under* $\boldsymbol{\mu}$ *starting from* $\mathbf{x}_{i,\bar{N}}$. *Then, for any* $i$,

$$O_i(\mathbf{x}^+) = O_i(\tilde{\mathbf{x}}^+). \tag{70}$$

*Proof.* We will prove this by showing that the states of all neighbors $\mathcal{N}_i(\mathbf{x}^+)$ agree. For each new neighbor $j \in \mathcal{N}_i(\mathbf{x}^+)$, since each agent can travel at most $\bar{d}$ in one step, this implies that

$$\|p_j - p_i\| \leq R + 2\bar{d}. \tag{71}$$

Applying Lemma 1 gives us that agent $j$'s observation is equal in both cases, i.e.,

$$O_j(\mathbf{x}) = O_j(\mathbf{x}_{i,\bar{N}}). \tag{72}$$

Hence, the controls $\mu(O_j(\mathbf{x}))$, and thus the new states match, i.e.,

$$x_j^+ = f_0(x_j, \mu(O_j(\mathbf{x}))) = f_0(x_j, \mu(O_j(\mathbf{x}_{i,\bar{N}}))) = \tilde{x}_j^+. \tag{73}$$

Since the new states for all agents in $\mathcal{N}_i(\mathbf{x}^+)$ agree, this implies that the new observation for agent $i$ must also agree. $\qquad\square$

We are now ready to prove Theorem A5.

*Proof of Theorem A5.* Let $\mathbf{x} \in \mathcal{X}_N$ such that $B(\mathbf{x}) \leq 0$. Then, applying Lemma 1 and Lemma 2 implies that the current and next observations for agent $i$ remain unchanged even when considering only the $\bar{N}$ closest agents, i.e.,

$$O_i(\mathbf{x}_{i,\bar{N}}) = O_i(\mathbf{x}), \tag{74}$$

$$O_i^+(\mathbf{x}_{i,\bar{N}}) = O_i^+(\mathbf{x}). \tag{75}$$

Hence, taking $\tilde{x} = \mathbf{x}_{i,\bar{N}} \in \mathcal{X}_{\bar{N}}$ and using (64),

$$\tilde{B}(O_i^+(\mathbf{x})) - \tilde{B}(O_i(\mathbf{x})) + \alpha(\tilde{B}(O_i(\mathbf{x}))) = \tilde{B}(O_i^+(\mathbf{x}_{i,\bar{N}})) - \tilde{B}(O_i(\mathbf{x}_{i,\bar{N}})) + \alpha(\tilde{B}(O_i(\mathbf{x}_{i,\bar{N}}))) \tag{76}$$

$$= \tilde{B}(O_i^+(\tilde{\mathbf{x}})) - \tilde{B}(O_i(\tilde{\mathbf{x}})) + \alpha(\tilde{B}(O_i(\tilde{\mathbf{x}}))) \tag{77}$$

$$\leq 0. \tag{78}$$

$\qquad\square$

Theorem A5 implies that finding a **single** DGCBF $\tilde{B}$ that satisfies the DGCBF condition for $\bar{N}$ agents enables the **same** DGCBF $\tilde{B}$ to *also* be applied to larger numbers of agents $N > \bar{N}$.

### A.8 Proof of Corollary 1

*Proof.* Since $\tilde{V}^{h^{(m)}, \boldsymbol{\mu}}(o_i^0) = V_i^{h^{(m)}, \boldsymbol{\mu}}(\mathbf{x}^0) := \max_{k \geq 0} h^{(m)}(o_i^k)$, applying Theorem 2 gives us that

$$\tilde{V}^{h^{(m)}, \boldsymbol{\mu}}(o_i^+) - \tilde{V}^{h^{(m)}, \boldsymbol{\mu}}(o_i) + \alpha\left( \tilde{V}^{h^{(m)}, \boldsymbol{\mu}}(o_i) \right) \tag{79}$$

$$= V_i^{h^{(m)}, \boldsymbol{\mu}}(\mathbf{x}^+) - V_i^{h^{(m)}, \boldsymbol{\mu}}(\mathbf{x}) + \alpha\left( V_i^{h^{(m)}, \boldsymbol{\mu}}(\mathbf{x}) \right) \tag{80}$$

$$\leq 0. \tag{81}$$

Thus, $\tilde{V}^{h^{(m)}, \boldsymbol{\mu}}$ is a DGCBF. $\qquad\square$

# B POLICY LOSS DETAILS

## B.1 DERIVATION OF (14)

We first start from (11), which we repeat below for convenience.

$$\min_{\theta} \quad \mathbb{E}_{\mathbf{x}\sim\rho_0,\mathbf{u}\sim\boldsymbol{\pi}_\theta(\cdot|\mathbf{x})}\left[Q^{\boldsymbol{\pi}_\theta}(\mathbf{x},\mathbf{u})\right], \tag{82a}$$

$$\text{s.t.} \quad \mathbb{E}_{\mathbf{x}\sim\rho^{\boldsymbol{\pi}_\theta}}\mathbb{E}_{\mathbf{u}\sim\boldsymbol{\pi}_\theta(\cdot|\mathbf{x})}\bigg[\underbrace{\underbrace{\max\left\{0,C^{(m)}(\mathbf{x},\mathbf{u})\right\}}_{:=\tilde{C}^{(m)}(\mathbf{x},\mathbf{u})}}_{:=\tilde{C}_\theta^{(m)}(\mathbf{x})}\bigg] \le 0, \quad \forall m. \tag{82b}$$

In the above, we additionally define $\tilde{C}_\theta^{(m)}(\mathbf{x}) := \max\{0, C^{(m)}(\mathbf{x},\mathbf{u})\}$ for convenience.

Borrowing the ideas of gradient projection from multi-objective optimization (Yu et al., 2020; Liu et al., 2021a), we combine the gradient from the objective minimization (82a), and the gradient from constraint satisfaction (82b) (equivalent to constraint minimization due to the constraints being non-negative) by projecting the gradient of (82a) such that it is orthogonal to the gradient of *all $m$* constraints (82b). We can do this in a **single** backward pass by using Informal Theorem 3.

Let $\psi$ denote the *stop gradient* function, such that the gradient of $\psi$ is equal to zero. Then, for a state distribution $\rho$,

$$\sigma := \mathbb{E}_{\mathbf{x}\sim\rho}[\nabla_\theta \max_m \tilde{C}_\theta^{(m)}(\mathbf{x})] \tag{83}$$

$$= \mathbb{E}_{\mathbf{x}\sim\rho}\mathbb{E}_{\mathbf{u}\sim\boldsymbol{\pi}_\theta(\mathbf{x})}\left[\nabla_\theta \log \boldsymbol{\pi}_\theta(\mathbf{x},\mathbf{u}) \max_m \tilde{C}^{(m)}(\mathbf{x},\mathbf{u})\right], \tag{84}$$

$$= \mathbb{E}_{\mathbf{x}\sim\psi(\rho)}\mathbb{E}_{\mathbf{u}\sim\psi(\boldsymbol{\pi}_\theta(\mathbf{x}))}\left[\nabla_\theta \log \boldsymbol{\pi}_\theta(\mathbf{x},\mathbf{u}) \max_m \tilde{C}^{(m)}(\mathbf{x},\mathbf{u})\right], \tag{85}$$

$$= \nabla_\theta \mathbb{E}_{\mathbf{x}\sim\psi(\rho)}\mathbb{E}_{\mathbf{u}\sim\psi(\boldsymbol{\pi}_\theta(\mathbf{x}))}\left[\log \boldsymbol{\pi}_\theta(\mathbf{x},\mathbf{u}) \max_m \tilde{C}^{(m)}(\mathbf{x},\mathbf{u})\right], \tag{86}$$

where the second line uses the score function gradient (Appendix A.4), and

$$g = \mathbb{E}_{\mathbf{x}\sim\rho^{\boldsymbol{\pi}_\theta}}\left[\mathbb{1}_{\{\max_m \tilde{C}_\theta^{(m)}(\mathbf{x})\le 0\}}\mathbb{E}_{\mathbf{u}\sim\boldsymbol{\pi}_\theta(\cdot|\mathbf{x})}\left[\nabla_\theta \log \boldsymbol{\pi}_\theta(\mathbf{x},\mathbf{u})Q^{\boldsymbol{\pi}_\theta}(\mathbf{x},\mathbf{u})\right]\right], \tag{87}$$

$$= \mathbb{E}_{\mathbf{x}\sim\psi(\rho^{\boldsymbol{\pi}_\theta})}\left[\mathbb{1}_{\{\max_m \tilde{C}_\theta^{(m)}(\mathbf{x})\le 0\}}\mathbb{E}_{\mathbf{u}\sim\psi(\boldsymbol{\pi}_\theta(\cdot|\mathbf{x}))}\left[\nabla_\theta \log \boldsymbol{\pi}_\theta(\mathbf{x},\mathbf{u})\psi(Q^{\boldsymbol{\pi}_\theta}(\mathbf{x},\mathbf{u}))\right]\right], \tag{88}$$

$$= \nabla_\theta\mathbb{E}_{\mathbf{x}\sim\psi(\rho^{\boldsymbol{\pi}_\theta})}\left[\mathbb{1}_{\{\max_m \tilde{C}_\theta^{(m)}(\mathbf{x})\le 0\}}\mathbb{E}_{\mathbf{u}\sim\psi(\boldsymbol{\pi}_\theta(\cdot|\mathbf{x}))}\left[\log \boldsymbol{\pi}_\theta(\mathbf{x},\mathbf{u})\psi(Q^{\boldsymbol{\pi}_\theta}(\mathbf{x},\mathbf{u}))\right]\right], \tag{89}$$

where the second line follows from the policy gradient theorem. Let $\tilde{g} := \nu\sigma^{(m)} + g$. , where we take the $\rho$ in the definition of $\sigma^{(m)}$ to be equal to $\rho^{\boldsymbol{\pi}_\theta}$ (we discuss the implications of this in the next subsection Appendix B.2):

**Theorem A6.** *The combined gradient $\tilde{g}$ can be computed as the gradient of the loss function $L$ defined in (14), i.e.,*

$$\tilde{g} = \nabla_\theta L(\theta), \tag{90}$$

*where*

$$L(\theta) := \mathbb{E}_{\mathbf{x}\sim\psi(\rho^{\boldsymbol{\pi}_\theta})}\mathbb{E}_{\mathbf{u}\sim\psi(\boldsymbol{\pi}_\theta(\cdot|\mathbf{x}))}\left[\log \boldsymbol{\pi}_\theta(\mathbf{x},\mathbf{u})\,\psi\big(\tilde{Q}(\mathbf{x},\mathbf{u},\theta)\big)\right], \tag{91}$$

$$\tilde{Q}(\mathbf{x},\mathbf{u},\theta) := \begin{cases} Q^{\boldsymbol{\pi}_\theta}(\mathbf{x},\mathbf{u}), & \tilde{C}_\theta^{(m)}(\mathbf{x}) \le 0, \ \forall m, \\ \nu \max_m \tilde{C}^{(m)}(\mathbf{x},\mathbf{u}), & \textit{otherwise.} \end{cases} \tag{92}$$

$$= \mathbb{1}_{\{\max_m \tilde{C}_\theta^{(m)}(\mathbf{x})\le 0\}}\psi(Q^{\boldsymbol{\pi}_\theta}(\mathbf{x},\mathbf{u})) + \nu \max_m \tilde{C}^{(m)}(\mathbf{x},\mathbf{u}). \tag{93}$$

*Proof.* Taking $\rho = \rho^{\pi_\theta}$ for $\sigma$ and summing the expressions for $\sigma$ and $g$ in (86) and (89) respectively gives

$$\nu\sigma + g = \nu\nabla_\theta \, \mathbb{E}_{\mathbf{x}\sim\psi(\rho^{\pi_\theta})} \, \mathbb{E}_{\mathbf{u}\sim\psi(\pi_\theta(\cdot|\mathbf{x}))} \left[ \log \pi_\theta(\mathbf{x},\mathbf{u}) \, \max_m \tilde{C}^{(m)}(\mathbf{x},\mathbf{u}) \right] \tag{94}$$
$$+ \nabla_\theta \mathbb{E}_{\mathbf{x}\sim\psi(\rho^{\pi_\theta})} \left[ \mathbb{1}_{\{\max_m \tilde{C}_\theta^{(m)}(\mathbf{x})\leq 0\}} \, \mathbb{E}_{\mathbf{u}\sim\psi(\pi_\theta(\cdot|\mathbf{x}))} \left[ \log \pi_\theta(\mathbf{x},\mathbf{u}) \psi(Q^{\pi_\theta}(\mathbf{x},\mathbf{u})) \right] \right]$$

$$= \nabla_\theta \, \mathbb{E}_{\mathbf{x}\sim\psi(\rho^{\pi_\theta})} \, \mathbb{E}_{\mathbf{u}\sim\psi(\pi_\theta(\cdot|\mathbf{x}))} \left[ \log \pi_\theta(\mathbf{x},\mathbf{u}) \left( \nu \max_m \tilde{C}^{(m)}(\mathbf{x},\mathbf{u}) + \psi(Q^{\pi_\theta}(\mathbf{x},\mathbf{u})) \right) \right] \tag{95}$$

$$= \nabla_\theta \, \mathbb{E}_{\mathbf{x}\sim\psi(\rho^{\pi_\theta})} \, \mathbb{E}_{\mathbf{u}\sim\psi(\pi_\theta(\cdot|\mathbf{x}))} \left[ \log \pi_\theta(\mathbf{x},\mathbf{u}) \psi\big( \underbrace{\nu \max_m \tilde{C}^{(m)}(\mathbf{x},\mathbf{u}) + Q^{\pi_\theta}(\mathbf{x},\mathbf{u})}_{:=\bar{Q}(\mathbf{x},\mathbf{u},\theta)} \big) \right] \tag{96}$$

$$= \nabla_\theta \, \mathbb{E}_{\mathbf{x}\sim\psi(\rho^{\pi_\theta})} \, \mathbb{E}_{\mathbf{u}\sim\psi(\pi_\theta(\cdot|\mathbf{x}))} \left[ \log \pi_\theta(\mathbf{x},\mathbf{u}) \psi\big( \bar{Q}(\mathbf{x},\mathbf{u},\theta) \big) \right] \tag{97}$$

$\square$

In other words, Theorem A6 implies that $\tilde{g}$ can be computed using a single backward pass.

### B.2 Implications of minimizing the constraint violation over $\rho^{\pi_\theta}$

Note that a key step that allows us to move the gradient operator between the inside (83) and outside (86) for the expression of $\sigma$ is our use of $\psi$, because this allows this equivalent holding even when we take $\rho = \rho^{\pi_\theta}$.

*Without* the $\psi$ and taking $\rho = \rho^{\pi_\theta}$, we would obtain that (86) is *instead* equivalent to

$$\nabla_\theta \mathbb{E}_{\mathbf{x}\sim\rho^{\pi_\theta}} \mathbb{E}_{\mathbf{u}\sim\pi_\theta(\cdot|\mathbf{x})} [\max_m \tilde{C}^{(m)}(\mathbf{x},\mathbf{u})] = \nabla_\theta \mathbb{E}_{\mathbf{x}_0\sim\rho_0, \mathbf{u}^k\sim\pi_\theta(\cdot|\mathbf{x}^k)} \left[ \underbrace{\sum_{k=0}^{\infty} \max_m \tilde{C}^{(m)}(\mathbf{x}^k,\mathbf{u}^k)}_{:=Q^{C,\pi_\theta}(\mathbf{x}_0,\mathbf{u})} \right] \tag{98}$$

$$\propto \mathbb{E}_{\mathbf{x}\sim\rho^{\pi_\theta}} \mathbb{E}_{\mathbf{u}\sim\pi_\theta(\cdot|\mathbf{x})} \left[ \nabla_\theta \log \pi_\theta(\mathbf{x},\mathbf{u}) \, Q^{C,\pi_\theta}(\mathbf{x},\mathbf{u}) \right], \tag{99}$$

$$\neq \mathbb{E}_{\mathbf{x}\sim\rho^{\pi_\theta}} \mathbb{E}_{\mathbf{u}\sim\pi_\theta(\mathbf{x})} \left[ \nabla_\theta \log \pi_\theta(\mathbf{x},\mathbf{u}) \, \max_m \tilde{C}^{(m)}(\mathbf{x},\mathbf{u}) \right]. \tag{100}$$

Note that the second line comes from the use of the **policy gradient theorem**, while the expression in (100) comes from the application of the **score function gradient** Appendix A.4. Consequently, while (100) will minimize the maximum DCBF violation $\tilde{C}^{(m)}$, (99) minimizes the *sum* of **future** DCBF violations as well.

Another way to see this is to note that (98) can be viewed as an optimal control problem with *cost* equal to the DCBF violation. In other words, in (98) the policy will additionally try to move to states where the DCBF violation is small as opposed to changing only the control $\mathbf{u}$ such that it satisfies the DCBF conditions in (100).

**Experimental Validation.** To validate our intuition above, we conduct experiments in the Transport2 environment to compare using (98) with (100), and plot the training curves in Figure 7. The results show that using (98) converges much slower in cost, which matches the intuition above. Namely, (98) *additionally* tries to avoid *states* where the DCBF constraint violation is high, which leads to unnecessary conservatism.

## C Experiments

### C.1 Computation resources

The experiments are run on a 13th Gen Intel(R) Core(TM) i7-13700KF CPU with 64GB RAM and an NVIDIA GeForce RTX 4090 GPU. The training time is around 12 hours for $2 \times 10^5$ steps for DGPPO, 14 hours for Lagr and 10 hours for Penalty.

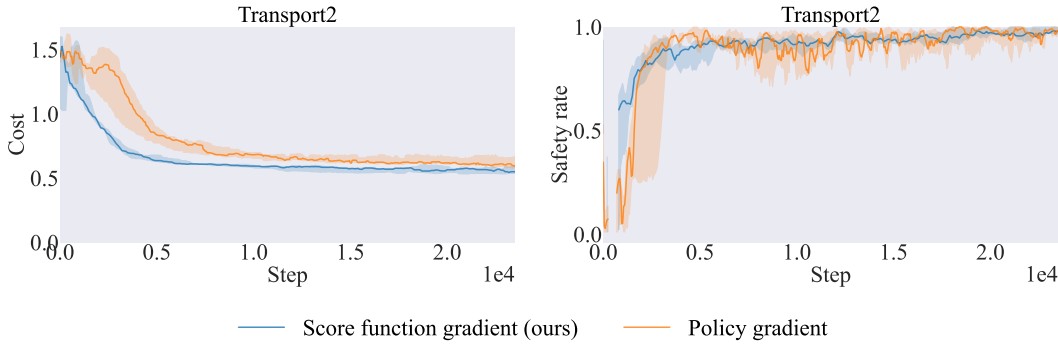

Figure 7: In comparison to DGPPO which uses (100), using (98) converge much slower in cost, which matches the intuition that (98) is optimizing for the wrong objective due to unnecessarily avoiding states where the DCBF constraint violation is high.

## C.2 ENVIRONMENTS

### C.2.1 LiDAR ENVIRONMENTS

In the LiDAR environments, we assume that the agents have a radius of $r = 0.05$ and a local sensing radius $R = 0.5$ such that one agent can observe other agents or obstacles only when they are within its sensing radius. Agents use LiDAR to detect obstacles. For each agent, there are 32 evenly-spaced LiDAR rays originating from each agent measuring the relative location of obstacles. To reduce the size of the multi-agent graph, the 8 shortest LiDAR rays are returned.

We use directed graphs $\mathcal{G} = (\mathcal{V}, \mathcal{E})$ to represent the LiDAR environments. $\mathcal{V}$ is the set of nodes containing the objects in the environments, including agents $\mathcal{V}_a$, goals/landmarks $V_g$, and the hitting points of LiDAR rays (obstacles) $\mathcal{V}_o$. The edges $\mathcal{E} \subseteq \{(i,j) \mid i \in \mathcal{V}_a, j \in \mathcal{V}\}$ denote the information flow from a sender node $j$ to a receiver node (agent) $i$. An edge $(i,j)$ exists only if the distance between node $i$ and $j$ are smaller than the sensing radius $R$. The neighbor nodes of agent $i$ is defined as $\mathcal{N}_i := \{j \mid (i,j) \in \mathcal{E}\}$, so that the information flow happens between the agents and their neighbors. The node features $v_i$ include the state of the node $x_i$ and a one-hot encoding of the type of the node $i$ (e.g., agent, goal/landmark, LiDAR hitting points). The edge features $e_{ij}$ include the information passed from node $j$ to node $i$, including the relative positions and velocities.

In all LiDAR environments, we include 3 rectangle-shaped obstacles, and the agents need to avoid inter-agent collision and agent-obstacle collisions.

We consider 4 LiDAR environments: TARGET, SPREAD, LINE, and BICYCLE:

**TARGET:** The agents need to reach their pre-assigned goals (Figure 2a).

**SPREAD:** The agents need to collectively cover a set of goals without having access to an assignment (Figure 2b).

**LINE:** The agents need to form a line between two given landmarks (Figure 2c).

**BICYCLE:** The agents follow the more difficult bicycle dynamics. The task here is the same as TARGET (Figure 2d).

The agents in the first 3 environments follow the double integrator dynamics. The state of agent $i$ is $x_i = [p_i^x, p_i^y, v_i^x, v_i^y]^\top$, where $[p_i^x, p_i^y]^\top := p_i \in \mathbb{R}^2$ is the position of agent $i$, and $[v_i^x, v_i^y]$ is its velocity. The control inputs are $u_i = [a_i^x, a_i^y]^\top$, which are the acceleration along the x-axis and y-axis. The agents follow the dynamics

$$\dot{x}_i = \begin{bmatrix} v_i^x & v_i^y & a_i^x & a_i^y \end{bmatrix}^\top. \tag{101}$$

We limit the control inputs of the agents to be within $[-1, 1]$, and also the velocities to be within $[-10, 10]$. In the BICYCLE environment, the state of agent $i$ is defined with

$x_i = [p_i^x, p_i^y, \cos\theta, \sin\theta, v]^\top$, where $\theta$ is the heading and $v$ is the speed. The control inputs are $u_i = [\delta, a]^\top$, including the steering angle $\delta$ and the acceleration $a$. The agents follow the bicycle dynamics given by

$$\dot{x}_i = [v\cos\theta \quad v\sin\theta \quad -v\sin\theta\tan\delta \quad v\cos\theta\tan\delta \quad a]^\top . \qquad (102)$$

The control inputs are limited by $v \in [-10, 10]$ and $\delta \in [-1.47, 1.47]$. In all LiDAR environments, we use a simulation time step of $0.03$ seconds and a total horizon of $128$ time steps. For all LiDAR environments, the edge features are defined as the relative positions and relative velocities between the nodes.

In LiDAR environments, the constraint function $h$ contains two parts: agent-agent collisions and agent-obstacle collisions. The agent-agent collisions $h^{(1)}$ function is defined as

$$h^{(1)}(o_i) = 2r - \min_{j \in \mathcal{N}_i} \|p_i - p_j\|, \qquad (103)$$

and the agent-obstacle collision $h^{(2)}$ function is

$$h^{(2)}(o_i) = r - \min_{j \in \mathcal{N}_i} \|p_i - p_j\|. \qquad (104)$$

For the cost functions $l$, we consider two types of them. The first type is used in the TARGET and the BICYCLE environments, where the agents need to *reach* their pre-assigned goals. We define this type of cost function with

$$l(\mathbf{x}, \mathbf{u}) = \frac{1}{N} \sum_{i=1}^{N} \left( 0.01\|p_i - p_i^{\text{goal}}\| + 0.001\text{sign}\left(\text{ReLU}(\|p_i - p_i^{\text{goal}}\| - 0.01)\right) + 0.0001\|u_i\|^2 \right),$$
$$(105)$$

where the first term penalizes the agents if they cannot reach the goal, the second term penalizes the agents if they cannot reach the goal exactly, and the third term encourages small controls. The second type of the cost functions is used in the SPREAD and the LINE environments, where the agents need to *cover* some goals/landmarks. We define this type of cost function with

$$l(\mathbf{x}, \mathbf{u}) = \frac{1}{N} \sum_{j=1}^{N} \min_{i \in \mathcal{V}_a} \left( 0.01\|p_i - p_j^{\text{goal}}\| + 0.001\text{sign}\left(\text{ReLU}(\|p_i - p_j^{\text{goal}}\| - 0.01)\right) \right.$$
$$\left. + 0.0001\|u_j\|^2 \right). \qquad (106)$$

Here, each goal finds its nearest agent and penalizes the whole team with the distance between them. In this way, the optimal policy of the agents is to cover all goals collaboratively.

### C.2.2 MuJoCo environments

For the TRANSPORT environment, we model the agents as double integrators and control the forces applied to each agent using the MuJoCo simulator (Todorov et al., 2012). We limit the control inputs of the agents to be within $[-1, 1]$. The agents need to collaboratively push a box from the inside so that the box reaches a given goal while avoiding colliding with each other.

We use a similar cost function $l(\mathbf{x}, \mathbf{u})$ as the TARGET environment but only for the box, i.e.,

$$l(\mathbf{x}, \mathbf{u}) = 0.01\|p_{\text{box}} - p_{\text{box}}^{\text{goal}}\| + 0.001\text{sign}\left(\text{ReLU}(\|p_{\text{box}} - p_{\text{box}}^{\text{goal}}\| - 0.01)\right) \qquad (107)$$

We usee a single constraint function $h^{(1)}$ for the agent-agent collision, defined as

$$h^{(1)}(o_i) = 2r - \min_{j \in \mathcal{N}_i} \|p_i - p_j\|, \qquad (108)$$

where $p_i$ denotes the position of agent $i$, and $r$ is the radius of the agent.

### C.2.3 VMAS environments

We use the REVERSETRANSPORT (which we call TRANSPORT2) and WHEEL environments from VMAS (Bettini et al., 2022; 2024). Note that the obstacles in the VMAS environments are not represented using LiDAR but using states including their positions and sizes. In both environments, the agents are modeled as double integrators and control the forces applied to each agent. We limit the control inputs of the agents to be within $[-1, 1]$.

**TRANSPORT2:**  In this environment, $N$ agents are placed in a square red package and must push the package from the inside to a goal location. The red package and the goal location are randomly sampled. Unlike the original REVERSETRANSPORT environment, we include the following two additional safety constraints:

- **Inter-agent collision avoidance:** The agents must avoid colliding with each other.
- **Package collision avoidance:** The center of the package must avoid colliding with three randomly placed circular obstacles.

We use a similar cost function $l(\mathbf{x}, \mathbf{u})$ as the TARGET environment but only for the package, i.e.,

$$l(\mathbf{x}, \mathbf{u}) = 0.01\|p_{\text{package}} - p_{\text{package}}^{\text{goal}}\| + 0.001\text{sign}\left(\text{ReLU}(\|p_{\text{package}} - p_{\text{package}}^{\text{goal}}\| - 0.01)\right) \quad (109)$$

We usee a two constraint functions $h^{(1)}$, $h^{(2)}$ for the agent-agent collision and package-obstacle collisions respectively. The agent-agent collision function $h^{(1)}$ is defined as

$$h^{(1)}(o_i) = 2r - \min_{j \in \mathcal{N}_i} \|p_i - p_j\|, \quad (110)$$

where $p_i$ denotes the position of agent $i$, and $r$ is the radius of the agent. The package-obstacle collision function $h^{(2)}$ is defined as

$$h^{(2)}(o_i) = r_{\text{obs}} - \min_{q \in \{1,2,3\}} \|p_{\text{package}} - p_q\|. \quad (111)$$

**WHEEL:**  In this environment, $N$ agents must collectively rotate a line anchored to the origin with a large mass. Unlike the original WHEEL environment, we modify the goal to be a target angle that the line must rotate to. We also include the following two additional safety constraints:

- **Inter-agent collision avoidance:** The agents must avoid colliding with each other.
- **Line collision avoidance:** The angle of the line must stay outside of a certain range of angles.

We use the same cost function $l(\mathbf{x}, \mathbf{u})$ as the TARGET environment but on the angle of the line, i.e.,

$$l(\mathbf{x}, \mathbf{u}) = \frac{1}{N} \sum_{i=1}^{N} \left(0.01\|p_i - p_i^{\text{goal}}\| + 0.001\text{sign}\left(\text{ReLU}(\|p_i - p_i^{\text{goal}}\| - 0.01)\right) + 0.0001\|u_i\|^2\right). \quad (112)$$

We usee a two constraint functions $h^{(1)}$, $h^{(2)}$ for the agent-agent collision and line-obstacle collisions respectively. The agent-agent collision function $h^{(1)}$ is defined as

$$h^{(1)}(o_i) = 2r - \min_{j \in \mathcal{N}_i} \|p_i - p_j\|, \quad (113)$$

where $p_i$ denotes the position of agent $i$, and $r$ is the radius of the agent. The package-obstacle collision function $h^{(2)}$ is defined as

$$h^{(2)}(o_i) = r_{\text{obs}} - |\theta_{\text{line}} - \theta_{\text{obs}}|, \quad (114)$$

where the absolute value on the angle $|\cdot|$ is defined as the minimum angle between the two angles.

### C.3 IMPLEMENT DETAILS AND HYPERPARAMETERS

In our experiment, we let all agents share the same parameters for their policies and value functions. More specifically, we parameterize the agent's policy with $\pi_\theta : \mathcal{O} \to \mathcal{U}$, cost-value function with $V_\phi^l : \mathcal{X} \to \mathbb{R}$, and constraint-value function $V_\psi^{h^{(m)}} : \mathcal{O} \to \mathbb{R}^m$ using graph transformers (Shi et al., 2020) with parameters $\theta$, $\phi$, and $\psi$, respectively. Since the cost-value function $V_\phi^l$ is centralized, after the graph transformer, we compute the average of all node features and pass that to a final layer (multi-layer perceptron) to obtain the global cost value for the MAS.

In Table 1, we provide the value of the common hyperparameters for DGPPO and the baselines. Besides these common hyperparameters, the value of the unique hyperparameters of DGPPO are provided in Table 2.

Table 1: Common hyperparameters of DGPPO and the baselines.

| Hyperparameter | Value | Hyperparameter | Value |
|---|---|---|---|
| policy GNN layers | 2 | RNN type | GRU |
| massage passing dimension | 32 | RNN data chunk length | 16 |
| GNN output dimension | 64 | RNN layers | 1 |
| number of attention heads | 3 | number of sampling environments | 128 |
| activation functions | ReLU | gradient clip norm | 2 |
| GNN head layers | (32, 32) | entropy coefficient | 0.01 |
| optimizer | Adam | GAE $\lambda$ | 0.95 |
| discount $\gamma$ | 0.99 | clip $\epsilon$ | 0.25 |
| policy learning rate | 3e-4 | PPO epoch | 1 |
| $V^l$ learning rate | 1e-3 | batch size | 16384 |
| network initialization | Orthogonal | layer normalization | True |
| $V^l$ GNN layers | 2 | | |

Table 2: Unique hyperparameters of DGPPO

| Hyperparameter | Value |
|---|---|
| $V^h$ GNN layers | 1 |
| $a$ | 0.3 |
| $\nu$ | Scheduled. Initialized at $\nu = 1$, then doubled at $0.5$ and $0.75$ of the total update steps. |

### C.4 IMPLEMENTATION OF THE BASELINES

We implement a JAX (Bradbury et al., 2018) version of the baselines following their original implementations:

- InforMARL: https://github.com/nsidn98/InforMARL (MIT license)

- MAPPO-L: https://github.com/chauncygu/Multi-Agent-Constrained-Policy-Optimisation (MIT License)

### C.5 TRAINING CURVES

Here, we provide the cost and safety rate during training for all algorithms in Figure 8. We can observe that DGPPO achieves stable training in all environments with only one constant set of hyperparameters.

### C.6 ADDITIONAL ABLATION STUDIES

#### C.6.1 COMPARISON BETWEEN THE DECOUPLING METHOD AND THE COUPLED METHODS

In Section 4.3, we have introduced a decoupling method to performance gradient descent update following Equation (14). Here, we empirically compare the decoupling method (14) with the CRPO-style coupling method (12) and the CRPO-style coupling method with the constraint being on $C(x, u)$ instead of $\max\{0, C(x, u)\}$ (Equation (10)). We conduct experiments in the TRANSPORT2 environment and compare the safety rate and cost of the converged policies with different methods in Figure 9. We can observe that DGPPO achieves a much lower cost compared with the coupling methods. This is because the coupling methods perform gradient descent to minimize $Q^{\pi_\theta}$ only when the *whole trajectory* is safe, i.e., $\mathbb{E}_{\mathbf{x} \sim \rho^{\pi_\theta}}[\tilde{C}_\theta(\mathbf{x})] \leq 0$. This is much more conservative than the safety requirement on the per-transition level used by the decoupling method, especially in the multi-agent case. Therefore, the coupling method has little chance to minimize $Q^{\pi_\theta}$ during training but focuses on safety, resulting in a safe policy with poor performance. On the other hand, if the coupling method is performed with $C(x, u)$ instead of $\max\{0, C(x, u)\}$ (Equation (10)), the learned policy is no longer safe. This matches our discussion in Section 4.3.

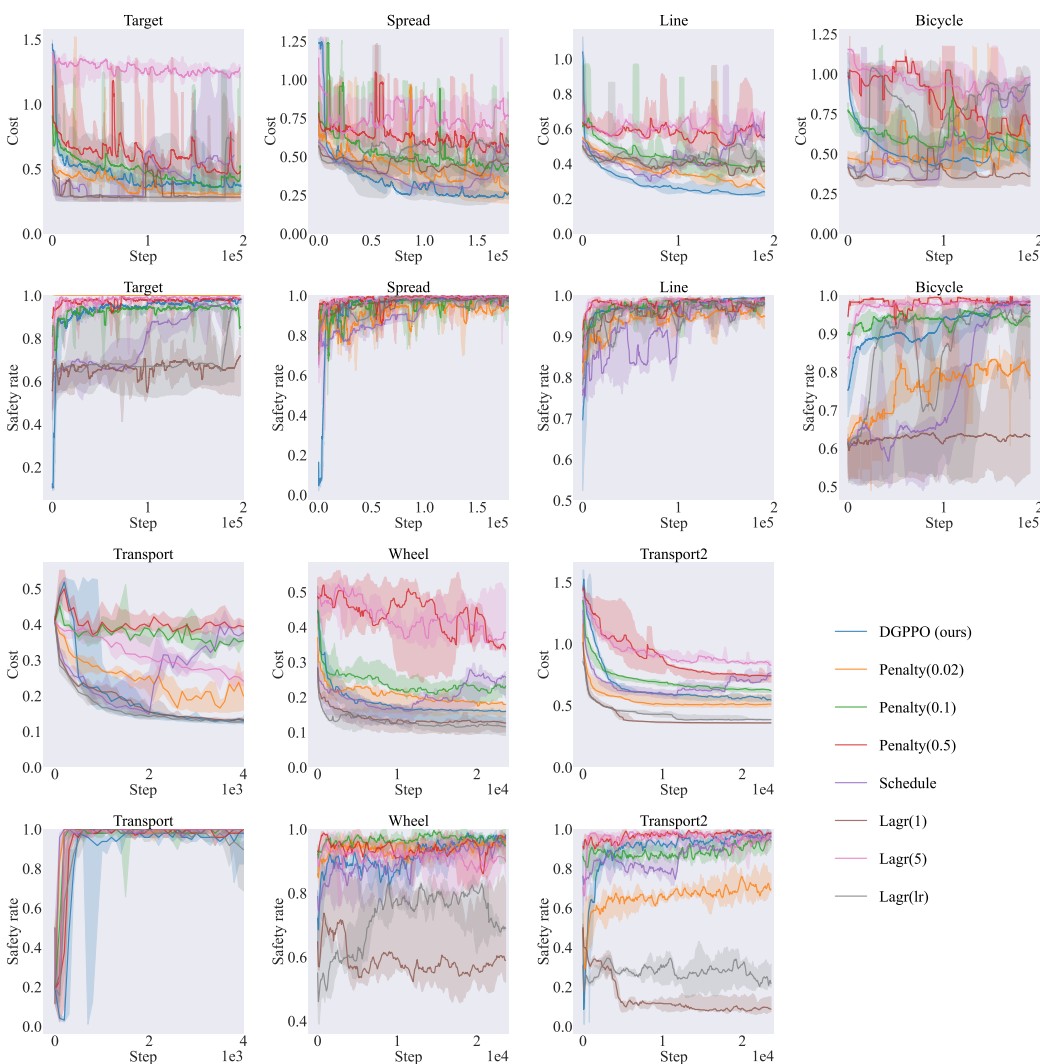

Figure 8: Costs and safety rates of DGPPO and the baselines during training.

### C.6.2 PROVIDE THE BASELINES WITH MORE DATA

In Section 4.5, we introduced that DGPPO requires sampling with both a stochastic and a deterministic policy. This means that DGPPO requires twice as much data per update step compared with the baselines, although the update steps are the same. Here, we answer the question that *what if the baselines are provided with double data*? We choose the WHEEL environment and select the three best baselines in Figure 3, namely Penalty(0.02), Penalty(0.1), and Schedule. To double the data, we consider two situations: doubling the batch size and keeping the training steps (similar to DGPPO), and doubling the training steps and keeping the batch size. The results are shown in Figure 10, where the semi-transparent colors show the original performance of the baselines, and the non-transparent colors show the performance of baselines after doubling the data. We can observe that after doubling the data, it is uncertain how the performance of the baselines changes. For example, Penalty(0.02) performs better than before with doubling the batch size in Figure 10a, but performs worse with doubling training steps in Figure 10b. On the contrary, DGPPO consistently outperforms all baselines.

### C.6.3 COMPARISON WITH CONSTRAINED OPTIMIZATION WITHOUT A CBF

As the proposed algorithm DGPPO is based on CBF, one natural question to ask is *what if we do not learn the CBF* but directly perform constrained optimization? To answer the question, we consider

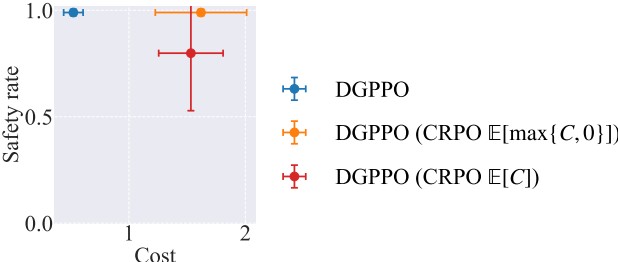

Figure 9: Comparison between DGPPO with the decoupling method and the CRPO-style update with $\mathbb{E}[\max\{0, C\}]$ and with $\mathbb{E}[C]$.

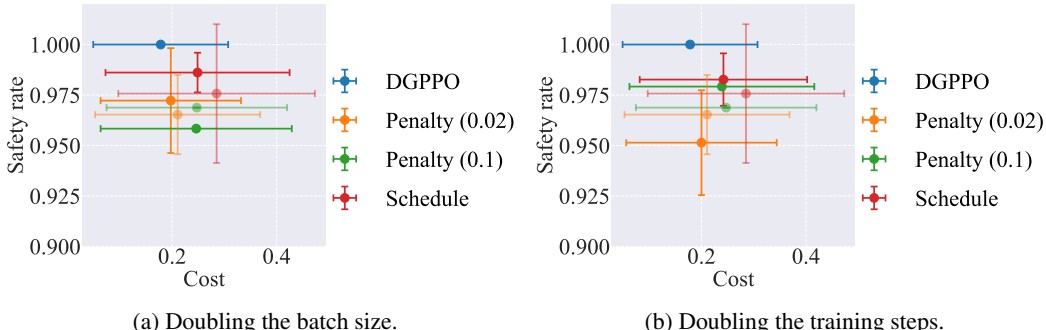

(a) Doubling the batch size.

(b) Doubling the training steps.

Figure 10: Costs and safety rates of DGPPO and three best baselines in the WHEEL environment. We also plot the original performance of the baselines in semi-transparent colors.

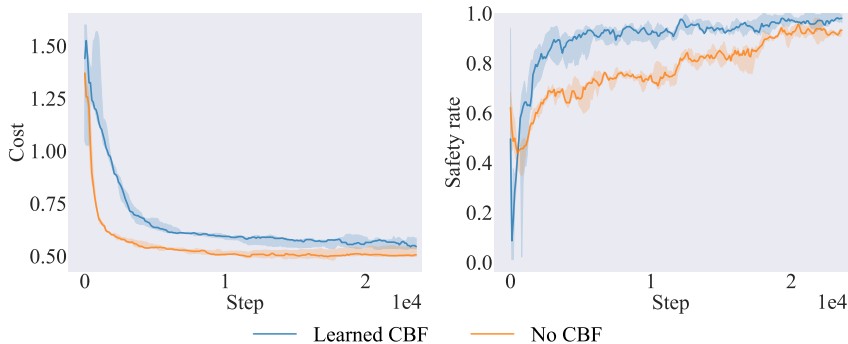

Figure 11: Comparison between DGPPO with doing constrained optimization without a CBF.

another baseline which changes the constraint in Equation (11b) to $\mathbb{E}_{\mathbf{x} \sim \rho^{\pi_\theta}}\left[\max\{0, h^{(m)}(\mathbf{x})\}\right] \leq 0$ while keep all other parts the same as DGPPO. We compare DGPPO (Learned CBF) with this new baseline (No CBF) in the TRANSPORT2 environment. The results are shown in Figure 11, which suggests that No CBF cannot achieve a safety rate that is as high as Learned CBF. Intuitively, it is because CBF not only constrains entering the avoid set, but also constrains the rate at which the agent can approach the safe-unsafe boundary. Therefore, it is more robust to estimation errors than directly doing constrained optimization.

### C.6.4 SENSITIVITY ANALYSIS OF $\nu$

In Section 5.3, we show that our proposed $\nu$ scheduling method achieves the best result compared with other choices of $\nu$. In particular, $\nu > 2$ results in slower convergence in the cumulative cost. Here, we further perform experiments to demonstrate the influence of $\nu$ on DGPPO. We consider the SPREAD environment with $\nu = 1, 2, 4, 6$, and train DGPPO until convergences. The results

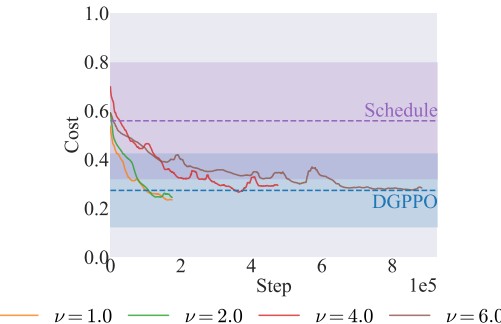

Figure 12: Influence of different $\nu$ on the convergence speed and the converged result of DGPPO. The dashed lines show the mean of DGPPO and the baseline with the best performance (Schedule), and the shades show the standard deviation.

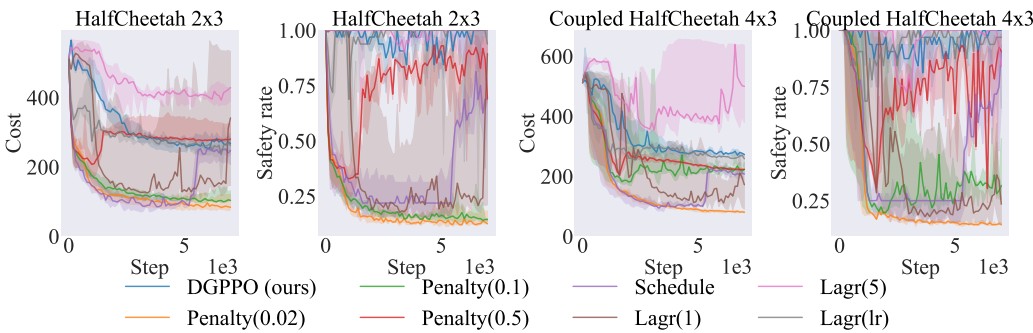

Figure 13: Cost and safety rates of DGPPO and the baselines during training in the Safe multi-agent MuJoCo environments.

are shown in Figure 12, where the mean cost and standard deviation of DGPPO and the baseline with the best performance (Schedule) are shown in dashed lines and shades. We can observe that although the convergence of DGPPO becomes slower with larger $\nu$, *the converged costs are the same*, and are much lower than Schedule. This phenomenon is different from the baselines, where different choices of hyperparameters directly affect the converged costs (See Figure 3).

### C.7 Additional environments in the safe multi-agent MuJoCo environments

To further demonstrate the ability of DGPPO in environments with complex discrete-time dynamics, here we consider another benchmark named safe multi-agent MuJoCo (Gu et al., 2023). We use the Safe HALFCHEETAH(2X3) and the Safe COUPLED HALFCHEETAH(4X3) tasks, where the agents control different subsets of joints of one or two cheetahs. The first number in the parenthesis denotes the number of agents, while the second number shows the number of joints that each agent controls. The agents need to work collaboratively to maximize the forward velocity but avoid colliding with a wall that moves forward at a predefined velocity. The results are shown in Figure 13, which suggests the same results as the main experiments in Section 5.2, that DGPPO has the best performance with a fixed set of hyperparameters.

### C.8 Scalability and Generalizability

Here we test the scalability and generalizability of DGPPO. Following Zhang et al. (2025), we define scalability as the number of agents during training, and generalizability as the ability to be deployed with more agents during test time.

Considering scalability, DGPPO has a similar performance as its based method GCBF+ (Zhang et al., 2025) as we have shown in Section 5.2 that DGPPO can be trained on 7 agents, and GCBF+ is trained with 8 in its original paper. In addition to Figure 5 in the main pages, here we also provide the comparison of the costs and safety rates of the converged policies of each algorithm in Figure 14.

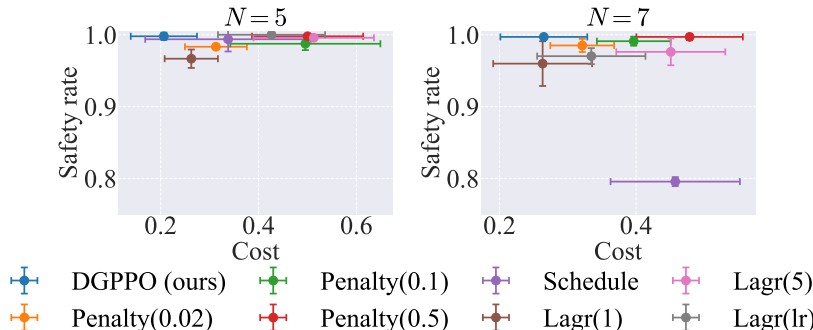

Figure 14: Costs and safety rates of the converged policies of DGPPO and the baselines in environments with $N = 5, 7$.

Table 3: Generalizability test results of DGPPO.

| Number of agents | Safety rate | Normalized cost |
|---|---|---|
| 8 | $1.000 \pm 0.000$ | $1.673 \pm 0.430$ |
| 16 | $0.992 \pm 0.088$ | $1.784 \pm 0.316$ |
| 32 | $0.987 \pm 0.112$ | $1.748 \pm 0.235$ |
| 64 | $0.986 \pm 0.118$ | $1.799 \pm 0.418$ |
| 128 | $0.982 \pm 0.133$ | $1.839 \pm 0.323$ |
| 256 | $0.985 \pm 0.122$ | $1.823 \pm 0.366$ |
| 512 | $0.985 \pm 0.123$ | $1.821 \pm 0.390$ |

Considering generalizability, GCBF+ can be deployed on $512$ agents without significant performance loss after training. Here, we perform a new experiment in the TARGET environment where DGPPO is also trained with $8$ agents. In Table 3, we show the test results of DGPPO deployed with larger numbers of agents. We observe that DGPPO maintains high safety rates and low costs when deployed on up to $512$ agents.

However, the above results are obtained in environments with the same agent density as the training environment. We cannot deploy DGPPO in environments with significantly higher agent density than the training environment because RL algorithms are sensitive to distribution shifts. We leave handling large distribution shifts to future work.

### C.9   CODE

The code of our algorithm and the baselines are provided in the 'dgppo.zip' file in the supplementary materials and online at `https://github.com/MIT-REALM/dgppo`.

## D   MORE DISCUSSION ABOUT DGPPO

### D.1   ADVANTAGES ON NOT DEPENDING ON A NOMINAL POLICY

As discussed in Section 1, one of the drawbacks of the CBF-based methods (Wang et al., 2017; Zhang et al., 2025) is that they require a nominal policy that can achieve high task performance. Here we further discuss why relying on a nominal performant policy is *not* a good idea.

**Requirement of Simple or Known Dynamics.**   Controller design usually requires the dynamics to be simple or requires knowledge of the dynamics. The PID controllers in Zhang et al. (2025) are constructed for the unicycle dynamics. More generally, PID controllers are usually only used with single-input single-output systems. For more complicated systems, one could use LQR or MPC, but this requires full knowledge of the dynamics. In addition, PID controllers are much more difficult to apply in environments with complex *contact dynamics*, for example, our TRANSPORT, WHEEL, and TRANSPORT2 environments.

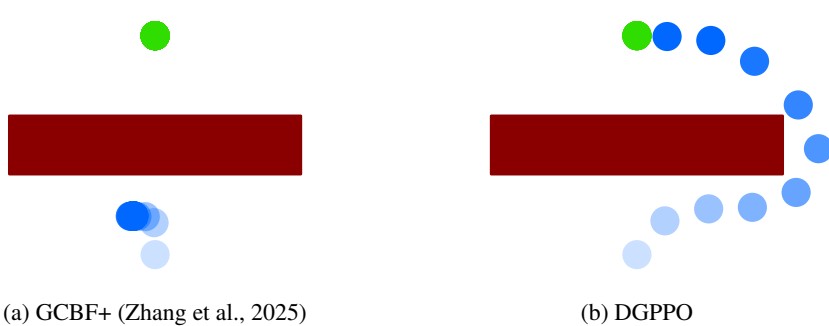

(a) GCBF+ (Zhang et al., 2025)          (b) DGPPO

Figure 15: Comparison of the converged policies learned using DGPPO and GCBF+ (Zhang et al., 2025) in a TARGET environment, where the agent (blue) needs to avoid the large obstacle (red) and reach the goal (green). The GCBF+ policy is myopic and gets stuck in a deadlock, while the DGPPO policy avoids the obstacle and reaches the goal to minimize the cumulative cost.

**Deadlocks.** Another drawback is that the CBF-QP approach of Zhang et al. (2025) leads to deadlocks, as discussed in Section VIII of Zhang et al. (2025) or theoretically in e.g. Grover et al. (2021). This is because the safety filter approach of Zhang et al. (2025) only minimizes deviations from the nominal policy at the current time step, even if this leads to a deadlock at a future time step. In contrast, minimizing the cumulative cost directly takes future behavior into account and hence will try to avoid deadlocks. Here we perform an additional experiment to demonstrate this. We apply the converged controller trained with GCBF+ and DGPPO respectively in the TARGET environment shown in Figure 15, where the agent needs to get around a large static obstacle and reach the goal. In the figure, we can observe that the GCBF+ policy gets in a deadlock behind the obstacle because the GCBF+ policy is myopic and only considers safety safe and minimizes the deviation from the reference controller at the *current* timestep. Therefore, stopping behind the obstacle is the optimal solution for GCBF+. On the contrary, the DGPPO policy successfully reaches the goal because it minimizes the *long-horizon* cumulative cost.

