# OpenReview forum: "Discrete GCBF Proximal Policy Optimization for Multi-agent Safe Optimal Control"
_ICLR.cc/2025/Conference — ICLR 2025 Poster_

### Official Review · Reviewer_LZax · 2024-10-23

**Soundness:** 2
**Presentation:** 2
**Contribution:** 2
**Rating:** 6
**Confidence:** 3

**Summary:**

This paper proposes a safe multi-agent reinforcement learning method based on distributed control barrier functions (CBFs) for multi-agent systems with limited perception capabilities. Simulation results on several multi-agent safe coordination tasks demonstrate the effectiveness of the proposed method.

**Strengths:**

(1) The multi-agent safe optimal control problem considered in this paper is both general and challenging, as neither the system model nor a nominal policy is available in advance.

(2) The learned policy is safe, which does not require additional safety filters in implementation.

(3) Extensive simulations are conducted, and state-of-the-art baselines are compared.

**Weaknesses:**

There exist several theoretical issues in the paper. The motivation of employing the discrete graph CBF is unclear. Some implementation details should be incorporated. See Questions part for more details.

**Questions:**

(1) The policy in this work only takes local observations as input, such that it is decentralized. Why do you refer to it as a distributed policy?

(2) The modified constraint (11b) is too strict which cannot be satisfied by a lot of policy classes, such as the Gaussian policy.

(3) In (12), please provide the explicit gradient formula when the safety condition is violated. Note that the authors provide a gradient in (41). Nevertheless, this gradient is not associated to the policy loss function (under safety violation) in (12).

(4) Theorem 3 is very difficult to understand. The orthogonality assumption is impractical. The reviewers also find that the authors try to replace the stationary state distribution (which is a function associated to the policy parameter $\theta$) with a constant state distribution in this theorem to obtain their gradient $g$ in (13). What is the purpose of giving Theorem 3? What is the relationship between (13) and (14)?

(5) The reason of using discrete graph CBFs should be explained clearly. Note that we can regard the multi-agent system as a large but single agent. Then, you can directly use the discrete CBF given in Theorem 2 to learn safe policies. In this case, the distributed control nature can still be preserved as the learned observation-based policy is end-to-end.

(6) Theorem 4 is hard to understand. What is the relationship between the discrete graph CBF and the discrete CBF? Similar to Theorem 1, it is important for the authors to show that the safe set is forward invariant based on the discrete graph CBF.

(7) In (11b), the safety constraint is calculated using a stochastic policy $\pi$. However, in Fig. 1, deterministic policies are used for estimating the discrete graph CBF.

(8) Why do the agents have different GAEs in your algorithm? Are you suggesting that the agents are heterogeneous and that their local policies differ?

---

> ### Author Response · Authors · 2024-11-23
> **Author Reply (1/4)**
>
> We thank the reviewer for finding our considered problem general and challenging and for acknowledging our extensive experiments. We also thank the reviewer for the detailed theoretical concerns from closely reading our manuscript.
>
> We hope that our responses below and our revised manuscript address the concerns raised by the reviewer.
>
> ## Summary
>
> In brief, we have:
>
> 1. Explained why the policy in this work is "distributed"
> 2. Clarified the difference between **policy gradients** and **score function gradients** during the policy updates w.r.t. the DCBF condition and performed additional experiments that demonstrate the difference
> 3. Provided proofs for the **safety** and **generalizability** guarantee of DGCBF.
> 4. Clarified the necessity of using deterministic rollouts to learn the constraint-value function $V^h$
> 5. Discussed the advantage of using DGCBF instead of DCBF
>
> ## Detailed Reply
>
> > **Q1:** The policy in this work only takes local observations as input, such that it is decentralized. Why do you refer to it as a distributed policy?
>
> **R**: Good question.
>
> We use the definition in this work that a distributed method allows for communication among agents (e.g., [1]), though we note that this term has a different meaning in other communities ([2]).
>
> Using this definition, whether the policy counts as distributed or not depends on whether the observation function $O_i$ uses communication or not.
> Since DGPPO is agnostic to both cases, we refer to it as a distributed policy.
>
> ---
>
> > **Q2:** The modified constraint (11b) is too strict which cannot be satisfied by a lot of policy classes, such as the Gaussian policy.
>
> **R**: Even if the modified constraint (11b) is too strict to be satisfied theoretically, **empirically** this results in a large increase in safety compared to the original more relaxed constraint (10) without significant differences in the cost (Figure 8 in App. B.6.1, Figure 9 in App. C.6.1 of the revised version).
>
> More importantly, this formulation enables the use of cheap gradient projections as in Theorem 3 to improve the sample efficiency, which outperforms both approaches above.
>
> While we are currently unable to explain this gap, we believe the empirical results are sufficient and leave stronger theoretical explanations of this phenomenon as future work.
>
> ---
>
> >**Q3:** In (12), when the safety condition is violated, the expression for the gradient, which requires the policy-gradient theorem, is not associated with the score function gradient in (41).
>
> **R:** Good catch, this is a typo. Thank you for the careful reading!
>
> Elaborating a bit more, there is a distinction between
>
>
> $$
> \nabla_\theta \mathbb{E}\_{\mathbf{x} \sim \rho^{\pi_\theta}}[ \tilde{C}\_\theta^{(m)}(\mathbf{x}) ] =
> \nabla\_\theta \mathbb{E}_\{\mathbf{x}\_0 \sim \rho\_0, \mathbf{u} \sim \pi\_\theta}\left[ \sum\_{k=0}^\infty  \max\big(0, \tilde{C}(\mathbf{x}^k, \mathbf{u}^k) \big) \right] \tag{$\star$}
> $$
>
>
> which requires the **policy-gradient theorem** to compute the gradient, and
>
>
> $$
> \nabla_\theta \mathbb{E}\_{\mathbf{x} \sim \rho}[ \tilde{C}\_\theta^{(m)}(\mathbf{x}) ] \Bigg|\_{\rho = \rho^{\pi_\theta}} = \mathbb{E}\_{\mathbf{x} \sim \rho^{\pi_\theta}}[ \nabla_\theta \tilde{C}\_\theta^{(m)}(\mathbf{x}) ] \tag{$\dagger$}
> $$
> which only uses the **score function gradient**.
>
> In particular, while our original goal is to satisfy the DCBF condition everywhere, i.e.,
> $$
> \tilde{C}_\theta^{(m)}(\mathbf{x}) \leq 0 \quad \forall \mathbf{x},
> $$
> ($\star$) will also try to minimize the stationary density at states $\mathbf{x}$ where the constraint is violated, which is **not** what we want.
> Hence, we use ($\dagger$) in (12) and in our experiments.
>
> We have included the above explanation in Appendix B.2 in the revised version.
>
> To verify this, **we have also performed additional experiments where we try using ($\star$) instead of ($\dagger$)**. The results show that using ($\star$) behaves overly conservatively and converges much slower in cost. This matches the discussion above, that using ($\star$) additionally tries to avoid _states_ where the DCBF constraint violation is high, which leads to unnecessary conservatism. We have included these new results in Appendix B.2 in the revised version.

---

> ### Author Response · Authors · 2024-11-23
> **Author Reply (2/4)**
>
> > **Q4 (1/4):** Theorem 3 is difficult to understand. What is the purpose of giving Theorem 3?
>
> **R:** Thanks for the feedback!
> Theorem 3 provides a way of computing an approximate *projected* objective function gradient ($g$) so that it does not interfere with the gradients from the constraint ($\sigma^{(m)}$).
>
> This allows us to borrow ideas from multi-objective optimization to improve the sample efficiency as compared to the CRPO-style of constrained optimization.
>
> To improve the clarity of Theorem 3, we have moved the formal statement of Theorem 3 into the Appendix (Theorem A2 in Appendix). Instead, we put an informal statement of Theorem 3 (Informal Theorem 3 in Section 4.3) in the revised version that more clearly communicates the above idea.
>
> Also, we have provided additional discussion on Theorem 3 in the revised manuscript (paragraph above (14)).
>
> ---
>
> > **Q4 (2/4):** The orthogonality assumption in Theorem 3 is impractical.
>
> **R:** While we do agree that the orthogonality assumption is impractical, it does allow us to derive a policy loss function that outperforms other alternatives (i.e., ablation study in Appendix B.6.1 of the original submission or Appendix C.6.1 of the revised version).
>
> Due to the unrealistic assumptions, we view Theorem 3 not as a core theoretical contribution but rather as a principled way to motivate design choices that behave well empirically, as showcased by our extensive empirical results.
>
> ---
>
> > **Q4 (3/4):** In Theorem 3, the expression for $\sigma$ uses a state distribution $\rho$ independent of $\theta$, but the expression for $g$ uses the stationary state distribution (which is a function of $\theta$).
>
> **R:** Good observation! The situation here is the same as in our response to **Q3**.
> Specifically, there is a difference between $\nabla_\theta \mathbb{E}\_{\mathbf{x} \sim \rho^{\pi_\theta}}$
> which requires the **policy-gradient theorem** to compute the gradient, and$\nabla_\theta \mathbb{E}\_{\mathbf{x} \sim \rho} \Bigg|\_{\rho = \rho^{\pi_\theta}}$, which uses the **score function gradient**.
>
> We have included a detailed discussion on this in Appendix B.2 in the revised version.
>
> ---
>
> > **Q4 (4/4):** What is the relationship between (13) and (14)?
>
> **R:** We realize that the derivation of the policy loss $L$ from Theorem 3 could be made clearer, and **have added a detailed derivation of the policy loss** to Appendix B in the revised version.
>
> Also, to make the connection more straightforward, we change the old (14) to use an indicator function instead of having two cases for the safe and unsafe cases, i.e.,
> $$
> \begin{align}
>     L(\theta)
>     &= \mathbb{E}\_{\mathbf{x} \sim \psi(\rho^{\pi_\theta})}
>     \mathbb{E}\_{\mathbf{u} \sim \psi(\pi\_\theta(\cdot|\mathbf{x}))}
>     \big[ \log \pi\_\theta(\mathbf{x}, \mathbf{u}) \psi\big( \tilde{Q}(\mathbf{x}, \mathbf{u}, \theta) \big) \big], \\\\
>     %%%%%%%
>     \tilde{Q}(\mathbf{x}, \mathbf{u}, \theta) &:=
>     \mathbb{1}\_{ \\{\max_m \tilde{C}^{(m)}\_\theta(\mathbf{x}) \leq 0 \\} } \psi(Q^{\pi_\theta}(\mathbf{x}, \mathbf{u})) + \nu \max\_m \tilde{C}^{(m)}(\mathbf{x}, \mathbf{u}).
> \end{align}
> $$

---

> > ### Author Response · Authors · 2024-11-23
> > **Author Reply (3/4)**
> >
> > > **Q5:** The reason for using discrete graph CBFs should be explained clearly. Note that we can regard the multi-agent system as a large but single agent. Then, you can directly use the discrete CBF given in Theorem 2 to learn safe policies. In this case, the distributed control nature can still be preserved as the learned observation-based policy is end-to-end.
> >
> > **R:** Thanks for the suggestion! The main reason is that a discrete graph CBF (DGCBF) provides an additional structure that **theoretically guarantees generalization to larger numbers of agents**.
> >
> > A DGCBF $\tilde{B}$ can be used to construct a DCBF $B$ (Appendix A.6 of the revised version). Hence, both can guarantee safety.
> > However, while $B$ only applies to a specific $N$, **we prove in Appendix A.7 of the revised version that a _single_ DGCBF $\tilde{B}$ that holds for $\bar{N}$ agents can be used to construct a DCBF for _any_ $N > \bar{N}$ agents**.
> > This provides theoretical evidence for training DGPPO on a small number of agents, then deploying it on a much larger number of agents.
> >
> > To further support this, we have performed experiments where DGPPO is trained with 8 agents and tested on up to 512 agents.
> > The results below suggest that DGPPO can maintain high performance even when deployed on more agents.
> >
> > |Number of agents|Safety rate|Normalized cost|
> > |---|---|---|
> > |8|$1.000\pm0.000$|$1.673\pm0.430$|
> > |16|$0.992\pm0.088$|$1.784\pm0.316$|
> > |32|$0.987\pm0.112$|$1.748\pm0.235$|
> > |64|$0.986\pm0.118$|$1.799\pm0.418$|
> > |128|$0.982\pm0.133$|$1.839\pm0.323$|
> > |256|$0.985\pm0.122$|$1.823\pm0.366$|
> > |512|$0.985\pm0.123$|$1.821\pm0.390$|
> >
> > We have added these additional experimental results in Appendix C.8 of the revised manuscript.
> >
> > ---
> >
> > > **Q6 (1/3):** Theorem 4 is hard to understand.
> >
> > **R:** Thanks for the feedback! Theorem 4 suggests that the attention mechanism can be used to construct a DGCBF $\tilde{B}$ such that the DGCBF condition (16) can be satisfied, even during neighborhood changes.
> >
> > To improve clarity, we have moved Theorem 4 to the Appendix.
> > Instead, we have replaced it with an informal version (Informal Theorem 4 in Section 4.4) in the revised paper that more clearly brings the key ideas across.
> >
> > ---
> >
> > > **Q6 (2/3):** What is the relationship between the discrete graph CBF and the discrete CBF?
> >
> > **R:** On one hand, a DGCBF $\tilde{B}$ can be used to construct a DCBF $B$ (Appendix A.6 of the revised version). Hence, both can guarantee safety.
> > On the other hand, while $B$ only applies to a specific $N$, **we prove in Appendix A.7 of the revised version that a _single_ DGCBF $\tilde{B}$ that holds for $\bar{N}$ agents can be used to construct DCBF for any $N > \bar{N}$**.
> >
> > ---
> >
> > > **Q6 (3/3):** Similar to Theorem 1, it is important for the authors to show that the safe set is forward invariant based on the discrete graph CBF.
> >
> > **R:** Thanks for the comment! We have proved **the safe set is forward invariant** using DGCBF in Appendix A.6 in the revised manuscript. Furthermore, we have also shown that DGCBF can be used to guarantee safety for **any number of agents** in Appendix A.7 in the revised manuscript.

---

> > > ### Author Response · Authors · 2024-11-23
> > > **Author Reply (4/4)**
> > >
> > > > **Q7:** In (11b), the safety constraint is calculated using a stochastic policy $\pi$. However, in Fig. 1, deterministic policies are used for estimating the discrete graph CBF.
> > >
> > > **R:** This observation is correct. DGPPO is **designed in this way** and it is not a mistake.
> > >
> > > While the CBF $V^{h^{(m)}}$ is **learned** using a deterministic policy, it is **evaluated** in the CBF constraint (11b) using a stochastic policy. Given a learned $V^{h^{(m)}}$ from a deterministic policy, the safety constraint in (11b) is calculated where the expression of $C^{(m)}$ (6b) depends on $B^{(m)} = V^{h^{(m)}}$.
> > >
> > > We do this because
> > > - $V^h$ is a DGCBF (Corollary 1), but the definition of the $V^h$ function requires a deterministic policy (Equation (7)).
> > > - Computing gradients with unknown dynamics using either the policy-gradient theorem or score function gradients requires a stochastic policy.
> > >
> > > We test the necessity of using a deterministic policy in the paragraph "Learning $V^h$ with a stochastic policy" in Section 5.3 of the original submission by using a stochastic policy to learn $V^h$ instead.
> > > However, this degrades both the cost and safety rate.
> > >
> > > ---
> > >
> > > > **Q8:** Why do the agents have different GAEs in your algorithm? Are you suggesting that the agents are heterogeneous and that their local policies differ?
> > >
> > > **R:** Thank you for the careful reading! This is a typo, and we have corrected this in the revised manuscript (Equation 21 in the revised manuscript), where the GAE (calculated from the cost) of all the agents are denoted as $A^\mathrm{GAE}$, and the advantage used for updating agent $i$'s policy is defined as $\tilde{A}_i$ (eighter equals to $A^\mathrm{GAE}$ or $\nu\max_m\hat{C}\_{\theta,i}^{(m)}$ depending on where the DGCBF constraint is satisfied).
> > >
> > > ## References
> > >
> > > [1] Garg, Kunal, et al. "Learning safe control for multi-robot systems: Methods, verification, and open challenges." Annual Reviews in Control 57 (2024): 100948.
> > >
> > > [2] Zhang, Kaiqing, et al. "Fully decentralized multi-agent reinforcement learning with networked agents." International conference on machine learning. PMLR, 2018.

---

> > > > ### Comment · Reviewer_LZax · 2024-11-27
> > > >
> > > > Thanks for your detailed answers. I have decided to raise my score.

---

> > > > > ### Author Response · Authors · 2024-11-27
> > > > > **Thank you very much for raising your score!**
> > > > >
> > > > > Thank you very much for raising your score! Your theoretical questions have greatly improved the theoretical rigor of our paper!
> > > > >
> > > > > If you have any further concerns that are holding you back from further raising your score, please let us know! We are more than happy to address them.

---

### Official Review · Reviewer_XeoF · 2024-11-02

**Soundness:** 2
**Presentation:** 3
**Contribution:** 2
**Rating:** 6
**Confidence:** 3

**Summary:**

The paper introduces a novel framework, Discrete Graph Control Barrier Functions Proximal Policy Optimization (DGPPO), for ensuring safe control in multi-agent systems (MAS) operating under unknown discrete-time dynamics and input constraints. Unlike prior approaches that rely on continuous-time models and known nominal policies, DGPPO incorporates discrete CBFs (DCBFs) and reinforcement learning to dynamically learn both a high-performance safe policy and a discrete graph CBF (DGCBF). Through extensive empirical validation across various simulated MAS environments, DGPPO claims to achieve both high task performance and safety without the need for hyperparameter adjustments specific to each environment.

**Strengths:**

The authors present an innovative combination of reinforcement learning and discrete-time CBFs to address the challenges of safety and task performance in MAS with unknown dynamics. The extension to the DCBF framework in discrete-time and the introduction of DGCBFs allow for neighborhood adaptability, overcoming limitations associated with continuous-time CBFs. The approach is well-motivated, tackling safety in unknown environments without requiring a predefined nominal policy—a substantial improvement for multi-agent reinforcement learning (MARL). I particularly appreciate the rigorous theoretical presentation to support the proposed approach.

**Weaknesses:**

While DGPPO introduces a novel safety mechanism for MAS, nonetheless I believe there are few critical concerns that could limit the effectiveness and general applicability of the approach.

**Lack of clear performance metrics**:

while DGPPO is shown to achieve high safety, the empirical results focus primarily on safety metrics and the minimization of constraints. It remains unclear if the agents are successfully accomplishing global objectives. Without metrics like mean global success rate or reward, it is difficult to assess if the agents are merely achieving safety (e.g., by staying stationary) rather than making meaningful progress toward the task goals while satisfying the safety constraints. This is especially relevant as the DGPPO framework does not incorporate a nominal policy, meaning that without these metrics, the experiments risks overlooking cases where the agents avoid unsafe states at the expense of task completion. Does the proposed framework present mechanisms to ensure that safety constraints do not excessively dominate the objective?

**Limited scalability experiments**:

the authors state that DGPPO scale well w.r.t to the baseline approach tested, however testing 5 agents instead of 3 as in the original experiment, I believe it is too limited to claim scalability of the proposed approach. Crucially, as stated from the authors themselves the proposed approach requires both stochastic and deterministic rollouts to enforce DGCBFs. While this approach ensures safety in discrete-time settings, it also introduces significant sample inefficiency, which may limit the framework’s scalability to larger or more complex MAS. Hence, an extensive test with for instance 10 or 15 agents would strength the results of the paper.
Moreover while the authors employ GNNs with attention mechanisms to handle changing agent neighborhoods, the computational complexity of GNNs in larger MAS could become important. In high-density environments with frequent neighborhood changes, maintaining an updated and accurate DGCBF through GNNs could pose significant computational challenges, possibly impacting real-time applicability. A detailed discussion on the scalability of GNN-based policies for larger agent systems would add valuable context to the method’s limitations.

**Dependence on hyperparameter $\nu$ for constraint enforcement**:

the authors claim on the fact that DGPPO is less sensitive to hyperparameters does not seem to be properly backed up. From the plot in Fig. 6b the value of $\nu$—responsible for controlling the weight on constraint minimization steps—significantly impacts performance. Misalignment in $\nu$ could lead to either overly conservative or unsafe policies, showing that DGPPO still requires careful tuning, contrary to its stated hyperparameter robustness.

**Questions:**

Q1: How does DGPPO ensure that agents achieve the global objective, rather than just meeting safety constraints?

Q2: Could DGPPO’s rollout requirements be reduced to improve sample efficiency without compromising safety?

Q3: What are the practical scalability limits of DGPPO when applied to larger MAS, particularly with the use of GNNs?

---

> ### Author Response · Authors · 2024-11-23
> **Author Reply (1/2)**
>
> We thank the reviewers for finding our approach innovative and well-motivated, and for acknowledging our theoretical presentations. **There are important misunderstandings that we hope to have clarified in both our revised manuscript and our response below**.
>
> We hope that our responses below and our revised manuscript address the concerns raised by the reviewer.
>
> ## Summary
>
> In brief, we have:
>
> 1. Clarified that the "cost" in our paper is the same as the "negative reward" in the CMDP setting
> 2. Provided additional experimental results showing that DGPPO can generalize to 512 agents
> 3. Provided experimental results to clarify that larger $\nu$ of DGPPO only affects the convergence rate and not the converged performance
> 4. Discussed the sampling efficiency of DGPPO
>
> ## Detailed Reply
>
> > **Weakness 1, Question 1 (1/2):** Without metrics like mean global success rate or reward, it is unclear if the agents are merely achieving safety or are successfully accomplishing the global objective.
>
> **R:** **We already report the global reward $\textbf{global reward} = - \textbf{global cost}$**. This may have been a misunderstanding.
>
> In our paper, the joint cost $l$ is task-oriented, equivalent to the negative reward, and **unrelated to the safety constraints**.
> This follows the convention in the optimal control community (e.g., [1]) and is **NOT** the cost in the CMDP setting.
>
> For example, in Appendix B.2.1 of our original submission (Appendix C.2.1 of the revision), we have provided detailed equations for the cost, which are mainly distances of the agents to their goals. Therefore, minimizing costs drives the agents to their goals.
>
> To prevent confusion, **we clarify that the cost $l$ is task-oriented and not safety-oriented** in Section 3.1 of the revised paper:
>
> - "Let the joint cost function $\text{describing the desired task}$ be denoted as $l: \mathcal{X} \times\mathcal U\to\mathbb R$"
>
> We have also added a footnote in the revised paper to explain the relationship with the CMDP setting:
>
> - "The cost function l here is **not** the _cost_ in CMDP. Rather, it corresponds to the _negative reward_ in CMDP."
>
> Consequently, **the results (e.g., Figure 3, 4, 5, 6 in Section 5) _do_ show metrics for accomplishing the global objective**.
> For example, in Figure 3, since achieving the objective (by minimizing the cost) and achieving safety can be conflicting objectives, some baseline methods have high task performance (i.e., low cost) but low safety rates, while some baseline methods have low task performance (i.e., high cost) but high safety rates.
>
> ---
>
> > **Weakness 1, Question 1 (2/2):** Does the proposed framework present mechanisms to ensure that safety constraints do not dominate the objective?
>
> **R:** **Yes**, but only if the safety constraints remain satisfied.
>
> This can be seen in Equation (20) or Figure 1 in the original submission:
> - At states $\mathbf{x}$ where the safety constraints are satisfied ($\tilde C^{(m)}(\mathbf{x}) \leq 0$), we use the gradient of the objective function
> - Otherwise, we use the gradient of the constraint
>
> Hence, if the safety constraints are all satisfied, then only the gradient of the objective function will be used.

---

> > ### Author Response · Authors · 2024-11-23
> > **Author Reply (2/2)**
> >
> > > **Weakness 2, Question 3 (1/3):** Does the need for both deterministic and stochastic rollouts limit the scalability of DGPPO to larger or more complex MAS?
> >
> > **R:** **No, this does not limit the scalability**.
> >
> > While the need for both deterministic and stochastic rollouts means that DGPPO requires twice the number of samples compared to baseline methods, we actually find **DGPPO takes similar or even less time than the baseline methods**.
> > In particular, DGPPO takes 12 hours, while Lagrangian methods take 14 hours.
> > This is because Lagrangian methods require an _additional_ backpropagation step to update the Lagrange multiplier, which dominates the additional time needed to perform the deterministic rollout.
> > However, the deterministic rollout still results in additional overhead compared to the baseline Penalty methods (10 hours).
> >
> > We have added the training times of the baselines to Appendix C.1 of the revised manuscript.
> >
> > ---
> >
> > > **Weakness 2, Question 3 (2/3):** An extensive test with for instance 10 or 15 agents would strengthen the results of the paper.
> >
> > Thank you for the suggestion! We actually go even further and **perform additional experiments where we keep increasing the number of agents until 512** and find that DGPPO can maintain high safety rates and low costs even on much larger MAS. We show the results in the table below.
> >
> > |Number of agents|Safety rate|Normalized cost|
> > |---|---|---|
> > |8|$1.000\pm0.000$|$1.673\pm0.430$|
> > |16|$0.992\pm0.088$|$1.784\pm0.316$|
> > |32|$0.987\pm0.112$|$1.748\pm0.235$|
> > |64|$0.986\pm0.118$|$1.799\pm0.418$|
> > |128|$0.982\pm0.133$|$1.839\pm0.323$|
> > |256|$0.985\pm0.122$|$1.823\pm0.366$|
> > |512|$0.985\pm0.123$|$1.821\pm0.390$|
> >
> > These results have been added to Appendix C.8 of the revised manuscript.
> >
> > ---
> >
> > > **Weakness 2, Question 3 (3/3):** Does the use of GNNs pose computational challenges with larger MAS?
> >
> > **R:** **No**. We have shown above that DGPPO can be applied to 512 agents. In addition, GNNs not only do **not** pose computational challenges with larger MAS, but also **enable** our algorithm to be applied to larger MAS because of its ability to handle changing neighborhoods, large numbers of neighbors, etc.
> >
> >
> > ---
> >
> > > **Weakness 3:** From Figure 6b, the performance of DGPPO seems to be sensitive to the weight $\nu$ of the constraint minimization step.
> >
> > **R:** This is a good observation.
> > However, we wish to make the following two points.
> >
> > #### 1. The performance of Converged Policy is Unaffected
> > Larger $\nu$ **only** affects the convergence rate of the cost and **not** the performance after convergence as long as $\nu>1$. In contrast, varying the hyperparameters of the baseline methods **directly affects the performance of the converged policy**.
> >
> > We have added **additional experiments to verify this experimentally** by running $\nu=4$ and $\nu=6$ for longer and observe that the performance of the _converged_ policy matches the performance using the $\nu$ schedule.
> > See Appendix C.6.4 and Figure 12 in the revised manuscript.
> >
> > #### 2. Insensitive to the _type_ of environment
> > Baseline methods are sensitive to **the type of environment**. Since the optimal hyperparameter changes for different environments, the baseline hyperparameters **do not generalize** across different environments.
> >
> > On the contrary, the fact that we use a single set of hyperparameters shows that our hyperparameters generalize across different environments, and DGPPO is not sensitive to the environment type.
> >
> > ---
> >
> > > **Question 2:** Could DGPPO’s rollout requirements be reduced to improve sample efficiency without compromising safety?
> >
> > **R:** As shown in paragraph "Learning $V^h$ with a stochastic policy" of Section 5.3 in the original submission,
> > while it is possible to use _stochastic_ instead of _deterministic_ rollouts, this degrades the cost and safety rate.
> > We thus choose to use both deterministic rollouts (to learn $V^h$, which is a DGCBF) and stochastic rollouts (to optimize the policy under unknown dynamics) to maximize the cost and safety rate of DGPPO.
> >
> > Note: we have shown in Appendix B.6.2 of the original submission (Appendix C.6.2 of the revised version) that, **even if the baselines are given double the number of samples**, they still do not achieve similar performance as DGPPO.
> >
> > Moreover, from a _computational_ perspective, doubling the sample requirement does not change the wall clock time compared to the baseline methods. This is because the computational bottleneck lies in the _backpropagation step_ and not the _sample collection_ step in our experiments.
> >
> > ## References
> >
> > [1] Chow, Yinlam, et al. "A lyapunov-based approach to safe reinforcement learning." NeurIPS 2018.

---

> > > ### Comment · Reviewer_XeoF · 2024-11-25
> > >
> > > Thanks for your answers that clarify most of the concerns I highlighted in the review.
> > > Based on this, I have decided to raise my score.

---

> > > > ### Author Response · Authors · 2024-11-25
> > > > **Thank you very much for raising your score!**
> > > >
> > > > Thank you very much for raising your score!
> > > >
> > > > Since your current score is borderline accept, **would you please let us know what concerns are holding you back from further raising the score?**
> > > > We are more than happy to address them.

---

### Official Review · Reviewer_uB5M · 2024-11-03

**Soundness:** 3
**Presentation:** 3
**Contribution:** 3
**Rating:** 8
**Confidence:** 4

**Summary:**

The proposed DGPPO framework addresses challenges in multi-agent systems (MAS) by learning both a discrete graph control barrier function (DGCBF) and a high-performance safe policy under unknown discrete-time dynamics, changing neighborhoods, and input constraints. DGPPO combines reinforcement learning and DGCBF, achieving high task performance and safety across varied environments without needing a pre-existing nominal policy or multiple hyperparameter sets, consistently outperforming other methods in both metrics of safety rate vs cost for various simulations.

**Strengths:**

- The proposed method (DGPPO) is an elegant way to solve the discrete-time distributed MASOCP (multi-agent safe optimal control problem) with unknown environment dynamics. This assumption was not present in previous work which had to differentiate through the given transition functions.
- The theorems introduced provide a solid foundation for the applicability of DGPPO in the discrete-time setting.

**Weaknesses:**

- Scalability appears limited (only up to 7 agents) compared to the continuous time setting of GCBF+ [1] (most likely due to the present of unknown environment dynamics and the noise introduced through the sample score function gradient).
- The stability of DGPPO compared to the baselines does not seem appropriately explained. Is it a purely empirical observation or is there some theoretical justification available?
- In the given setting, why is the assumption of unknown dynamics interesting? To me, the environments considered are purely the settings of [1] without using the environment dynamics directly (even though they are available). Would it not be a better idea to consider an environment where the dynamics are not as simple as the ones in [1] or some complex unknown function (for e.g., common Mujoco robots)?

References:

[1] GCBF+: A Neural Graph Control Barrier Function Framework for Distributed Safe Multi-Agent Control, Zhang et al, T-RO, 2024

**Questions:**

1. Why is the dependency of the algorithm on a nominal policy a bad idea in the given settings? Since it appears easy enough to construct one (say a PID controller like in [1]) for the environments given, is this the right direction?
2. What is the difference between the training and inference situations in terms of the number of agents? Does the algorithm need to be retrained for every new number of agents unlike in [1] where the algorithm was trained on 8 agents and deployed on up to 1024 agents (albeit being purely concerned with single goal reaching while avoiding collisions)?
3. With regards to the sample efficiency and computation requirements, how is DGPPO w.r.t. the baselines (I noticed the training time was listed as 12 hours on the reference specifications)? On a related note, how is the benefit of a constant set of hyperparameters demonstrated? Can we confidently say the hyperparameter search for the baselines takes significantly longer (in wall clock time on a comparable machine)?
4.What are the restrictions on the definition of the avoid set $\mathcal{A}_i$ and the assumptions on the function $h_i^{(m)}$? Do the avoid sets primarily represent distance to $y^k$ greater than some safe radius?
5. The LiDAR part of the observation appears less clear. From the appendix (Sec B.2.1) is it right to say that only the LiDAR environments use the 32 equally spaced ray capturing relative positions? How are the obstacles in the VMAS environments represented to the agent?
6. The experiments with scalability to multiple agents (Fig. 5) appear quite close to the baselines. Is there a better comparison available?

---

> ### Author Response · Authors · 2024-11-23
> **Author Reply (1/3)**
>
> We thank the reviewer for acknowledging our DGPPO as "an elegant way to solve the discrete-time distributed MASOCP (multi-agent safe optimal control problem) with unknown environment dynamics", and for acknowledging our theoretical analysis.
> We hope that our responses below and our revised manuscript address the concerns raised by the reviewer.
>
> ## Summary
>
> In brief, we have:
>
> 1. Explained that designing a nominal policy requires simple or known dynamics and can lead to deadlocks;
> 2. Added more experiments to show that the DGPPO policy also generalizes to more agents (512) after training;
> 3. Clarified some details regarding DGPPO including the sampling efficiency, hyperparameter sensitivity, and the definition of the avoid set;
> 4. Clarified the details of the environments in our experiments and the results using more figures;
> 5. Explained the difference in dynamics considered in our paper compared with [1].
>
> ## Detailed Reply
>
> > **Question 1:** Why is the dependency of the algorithm on a nominal policy a bad idea in the given setting? Since it appears easy enough to construct one (say a PID controller like in [1]) for the environments given, is this the right direction?
>
> **R:** Good question! There are 2 reasons why depending on a nominal policy is a drawback.
>
> #### 1. Requirement of Simple or Known Dynamics
> Controller design usually requires the dynamics to be simple or requires knowledge of the dynamics.
> The PID controllers in [1] are constructed for the unicycle dynamics. More generally, PID controllers are usually only used with single-input single-output systems. For more complicated systems, one could use LQR or MPC, but this requires **full knowledge** of the dynamics. In addition, PID controllers are much more difficult to apply in environments with complex **contact dynamics**, for example, our `Transport`, `Wheel`, and `Transport2` environments.
>
> #### 2. Deadlocks
> Another drawback is that the CBF-QP approach of [1] leads to deadlocks, as discussed in Section VIII of [1] or theoretically in e.g. [2].
> This is because the safety filter approach of [1] only minimizes deviations from the nominal policy at the current time step, even if this leads to a deadlock at a future time step.
>
> In contrast, minimizing the cumulative cost directly takes future behavior into account and hence will try to avoid deadlocks.
> To investigate this, **we have performed an additional experiment in Appendix D.1 in the revised manuscript**, where the approach from [1] results in a deadlock while our method successfully completes the task (See Figure 15 in the revised manuscript).
>
> We have added the above discussion on the limitations of assuming a nominal policy in Appendix D.1 of the revised manuscript.
>
> ---
>
> > **Question 2, Weakness 1:** Scalability appears limited (only up to 7 agents) compared to the continuous time setting of GCBF+ [1]. Can the algorithm generalize to larger numbers of agents? Does the algorithm need to be retrained for every new number of agents?
>
> **R:** Good observation! We wish to make a distinction between the **scalability** (number of agents during training) and **generalizability** (ability to be deployed with more agents during test time).
>
> Scalability-wise, GCBF+ also _trains_ on 8 agents (Section VI of [1]), which is similar to the number of agents considered in our training.
> Generalizability-wise, GCBF+ can be deployed on 512 agents without significant performance loss after training.
>
> We have performed new experiments and find that **DGPPO is also able to generalize and be deployed on 512 agents** after being trained on 8 agents. The following table shows the safety rates and the normalized cost (w.r.t. traveling distance and number of agents). We can observe that DGPPO maintains high safety rates and low costs when deployed on up to 512 agents.
>
> |Number of agents|Safety rate|Normalized cost|
> |---|---|---|
> |8|$1.000\pm0.000$|$1.673\pm0.430$|
> |16|$0.992\pm0.088$|$1.784\pm0.316$|
> |32|$0.987\pm0.112$|$1.748\pm0.235$|
> |64|$0.986\pm0.118$|$1.799\pm0.418$|
> |128|$0.982\pm0.133$|$1.839\pm0.323$|
> |256|$0.985\pm0.122$|$1.823\pm0.366$|
> |512|$0.985\pm0.123$|$1.821\pm0.390$|
>
> From the results above, we can conclude that DGPPO does **not** need to be retrained when deploying for larger numbers of agents.
>
> We have added the above results in Appendix C.8 of the revised manuscript.

---

> > ### Author Response · Authors · 2024-11-23
> > **Author Reply (2/3)**
> >
> > > **Question 3 (1/4):** With regards to the sample efficiency and computation requirements, how is DGPPO w.r.t. the baselines (I noticed the training time was listed as 12 hours on the reference specifications)?
> >
> > **R:** Although DGPPO introduces a deterministic rollout which doubles the number of environment samples for each update step, we have validated that the performance improvements of DGPPO are not due to the additional sample use in Appendix B.6.2 of the original version (C.6.2 of the revised version) by **giving the baseline methods double the sample budget**.
> >
> > In terms of computation time, we find that DGPPO **runs faster** (12 hours) compared to Lagrangian methods (14 hours). This is because Lagrangian methods require an _additional_ backpropagation step to update the Lagrange multiplier which seems to dominate the additional time needed to perform the deterministic rollout.
> > However, the deterministic rollout still results in additional overhead compared to the baseline Penalty methods (10 hours).
> >
> > We have added the training times of the baselines to Appendix C.1 of the revised manuscript.
> >
> > ---
> >
> > > **Question 3 (2/4):** On a related note, how is the benefit of a constant set of hyperparameters demonstrated? Can we confidently say the hyperparameter search for the baselines takes significantly longer (in wall clock time on a comparable machine)?
> >
> > **R:** The benefit of a constant set of hyperparameters is discussed in Paragraph "(Q1): DGPPO has the best performance and is hyperparameter insensitive." in Section 5.2 of the original submission.
> >
> > We observe that for fixed hyperparameters, the performance of baseline methods varies greatly when the environment changes. Consequently, changes in the environment require the hyperparameters to be fine-tuned again.
> >
> > Assuming a budget of 10 runs for hyperparameter optimization for the baseline methods and a training time of 10-14 hours per run, this implies a total time of 100-140 hours. This is **significantly longer** than our method which runs in only 12 hours.
> >
> > ---
> >
> > > **Question 3 (3/4):** What are the restrictions on the definition of the avoid set and the assumptions on the function?
> >
> > **R:** The only requirement is that the safety specification function $h_i$ can be written as a function of agent $i$'s **local observation** (Line 127 in original submission).
> >
> > ---
> >
> > > **Question 3 (4/4):** Do the avoid sets primarily represent distance greater than some safe radius?
> >
> > Many of our environments happen to have avoid sets defined using distance in the position space simply due to distance-based safety being common.
> > Using distance-based avoid sets is **not** a restriction.
> > For example, the avoid set of the `Wheel` environment is defined using angular distance, i.e., the angle of the wheel cannot be within some region.
> >
> > ---
> >
> > > **Question 4:** Is LiDAR only used in the LiDAR environments? How are the obstacles in the VMAS environments represented to the agents?
> >
> > **R:** **Yes**, only the LiDAR environments use LiDAR. The obstacles in the VMAS environments are represented using states including their positions and sizes following the original code [2]. We have clarified this in Appendix C.2.3 of the revised manuscript.
> >
> > ---
> >
> > > **Question 5:** The experiments with scalability to multiple agents (Fig. 5) appear quite close to the baselines. Is there a better comparison available?
> >
> > **R:** Thanks for the suggestion! To provide a clearer comparison, we **add a new plot** (Figure 14 in the appendix of the revised manuscript) that compares the cost and safety rate of the converged policies and observe that **DGPPO achieves a safety rate of near $100\%$, similar to the most conservative baselines, while achieving half of their cost**.

---

> > > ### Author Response · Authors · 2024-11-23
> > > **Author Reply (3/3)**
> > >
> > > > **Weakness 2:** The stability of DGPPO compared to the baselines does not seem appropriately explained. Is it a purely empirical observation or is there some theoretical justification available?
> > >
> > > **R:** The improvements to training stability are mostly from empirical observation.
> > >
> > > However, this does agree with prior literature which has pointed out the instability of Lagrangian-based methods in the _zero_ constraint threshold setting [A, B] and the improved training stability of purely primal methods such as CRPO (which we use) [C, D, E].
> > >
> > > ---
> > >
> > > > **Weakness 3:** Why is the assumption of unknown dynamics interesting? The environments used seem to be the same as in [1]. It would be a better idea to consider environments where the dynamics are more complicated than [1] (e.g., common MuJoCo robots).
> > >
> > > **R:** We respectfully disagree with the reviewer for the following reasons:
> > >
> > > 1. The environments used are **not** the same as in [1]. [1] considers only double integrator dynamics and unicycle dynamics, while we also consider the bicycle dynamics (`Bicycle` environment) and contact dynamics (`Transport`, `Wheel`, `Transport2` environments).
> > > 2. We **already** consider environments with **more complicated dynamcs**. For example, the `Transport`, `Wheel`, `Transport2` environments use the MuJoCo and VMAS simulators for **contact dynamics** and are not used in [1]. Contact dynamics are a big reason why it is important to handle unknown discrete-time dynamics due to their **discrete** nature, which prevents the application of continuous-time methods such as [1].
> > >
> > > However, we **agree** that considering more complicated dynamics can strengthen our empirical evaluation.
> > > **We have added new experiments on the HalfCheetah dynamics from the Safe Multi-agent MuJoCo benchmark [4] to the revised manuscript** in Appendix C.7.
> > >
> > > ## References
> > >
> > > [1] GCBF+: A Neural Graph Control Barrier Function Framework for Distributed Safe Multi-Agent Control, Zhang et al, T-RO, 2024.
> > >
> > > [2] Grover, Jaskaran Singh, Changliu Liu, and Katia Sycara. "Deadlock analysis and resolution for multi-robot systems." Algorithmic Foundations of Robotics XIV: Proceedings of the Fourteenth Workshop on the Algorithmic Foundations of Robotics 14. Springer International Publishing, 2021.
> > >
> > > [3] Matteo Bettini, Amanda Prorok, and Vincent Moens. Benchmarl: Benchmarking multi-agent reinforcement learning. Journal of Machine Learning Research, 25(217):1–10, 2024.
> > >
> > > [4] Shangding Gu, Jakub Grudzien Kuba, Yuanpei Chen, Yali Du, Long Yang, Alois Knoll, and Yaodong Yang. Safe multi-agent reinforcement learning for multi-robot control. Artificial Intelligence, 319:103905, 2023.
> > >
> > > [A] Mario Zanon and Sébastien Gros. Safe reinforcement learning using robust mpc. IEEE Transactions on Automatic Control, 66(8):3638–3652, 2020.
> > >
> > > [B] Tairan He, Weiye Zhao, and Changliu Liu. Autocost: Evolving intrinsic cost for zero-violation reinforcement learning. In Proceedings of the AAAI Conference on Artificial Intelligence, volume 37, pp. 14847–14855, 2023.
> > >
> > > [C] Tengyu Xu, Yingbin Liang, and Guanghui Lan. Crpo: A new approach for safe reinforcement
> > > learning with convergence guarantee. In International Conference on Machine Learning, pp.
> > > 11480–11491. PMLR, 2021.
> > >
> > > [D] Guan, Jiayi, et al. "POCE: Primal Policy Optimization with Conservative Estimation for Multi-constraint Offline Reinforcement Learning." Proceedings of the IEEE/CVF Conference on Computer Vision and Pattern Recognition. 2024.
> > >
> > > [E] Gu, Shangding, et al. "Balance reward and safety optimization for safe reinforcement learning: A perspective of gradient manipulation." Proceedings of the AAAI Conference on Artificial Intelligence. Vol. 38. No. 19. 2024.

---

> > > > ### Comment · Reviewer_uB5M · 2024-11-27
> > > > **Thanks for the clarifications and detailed response**
> > > >
> > > > I appreciate the detailed response provided by the authors and the additional clarifications (such as generalizability and scalability which I had overlooked). I have increased my score appropriately.

---

> > > > > ### Author Response · Authors · 2024-11-28
> > > > > **Thank you very much for increasing your score!**
> > > > >
> > > > > Thank you very much for increasing your score! Your valuable questions have greatly improved the presentation and clarity of our paper!

---

### Author Response · Authors · 2024-11-23
**Reply to all**

We thank the reviewers for their valuable comments.
We are excited that the reviewers have identified the importance of the problem ($\color{#648FFF}{\textsf{uB5M}}$, $\color{#DC267F}{\textsf{LZax}}$),
the novelty of our technical contributions (**all** reviewers),
appreciated our extensive empirical validation ($\color{#E69F00}{\textsf{XeoF}}$, $\color{#DC267F}{\textsf{LZax}}$),
and found the paper well-motivated with good presentation ($\color{#648FFF}{\textsf{uB5M}}$, $\color{#E69F00}{\textsf{XeoF}}$).
We believe that DGPPO takes a significant step towards greatly improving the safety of multi-agent systems with **unknown, discrete-time dynamics _without_ an available performant reference controller** by applying distributed graph CBFs and multi-agent reinforcement learning.

---

As _all_ reviewers have recognized our technical novelty, the primary concerns stem from the scalability and generalizability (all reviewers),
hyperparameter sensitivity ($\color{#E69F00}{\textsf{XeoF}}$),
and clarity of theoretical results ($\color{#DC267F}{\textsf{LZax}}$).

In our updated revision, we provide major improvements by clarifying _all_ raised questions.
We provide a brief summary of notable changes below. All references to sections refer to the **revised** version.

## 1. Generalizability of DGPPO

We have **added an additional theorem** in Appendix A.7 which theoretically proves that the discrete graph control barrier function (DGCBF) can be generalized to more agents than initially trained for.

Next, we **perform an additional experiment where we train DGPPO with 8 agents and deploy the learned policy on up to 512 agents** (Appendix C.8). The results, shown below, demonstrate that DGPPO can maintain a high safety rate and low costs even with larger numbers of agents.

|Number of agents|Safety rate|Normalized cost|
|---|---|---|
|8|$1.000\pm0.000$|$1.673\pm0.430$|
|16|$0.992\pm0.088$|$1.784\pm0.316$|
|32|$0.987\pm0.112$|$1.748\pm0.235$|
|64|$0.986\pm0.118$|$1.799\pm0.418$|
|128|$0.982\pm0.133$|$1.839\pm0.323$|
|256|$0.985\pm0.122$|$1.823\pm0.366$|
|512|$0.985\pm0.123$|$1.821\pm0.390$|

## 2. Hyperparameter Sensitivity

Figure 6(b) shows that using larger values of $\nu$ results in a slower reduction of the cumulative cost.
We have **performed additional experiments that confirm that only the rate and _not_ the final converged policy is affected by $\nu$** (Appendix C.6.4).
This is a _significant improvement_ compared to the baseline methods, where both the rate **and** the performance of the converged policy are greatly affected by the choice of hyperparameter.

## 3. More Clarification on Theoretical Analysis

We have significantly clarified our theoretical analysis. In short, we have:

1. Included a more detailed derivation of our proposed policy loss (Appendix B.1)
1. Clarified the difference between **policy gradients** and **score function gradients** during the policy updates w.r.t. the DCBF condition and performed additional experiments that demonstrate the difference (Appendix B.2)
2. Provided proofs for the **safety** (Appendix A.6) and **generalizability** (Appendix A.7) guarantee of DGCBF.

---

We hope the new presentation better presents the contributions of our method in improving the safety and task-performance of multi-agent systems by generalizing distributed CBFs to more general settings.

We have tried our best to resolve all the questions raised in the individual responses below.
If the reviewers have any additional questions/comments/concerns, please let us know.
We appreciate the reviewer's precious time in providing their valuable feedback.

---

### Meta-Review · Area_Chair_YiAy · 2024-12-21

**Metareview:**

Discrete GCBF Proximal Policy Optimization for Multi-agent Safe Optimal Control


Summary:  This paper proposes DGPPO, which integrates discrete graph control barrier functions (DGCBFs) with reinforcement learning to address the safety and performance challenges in multi-agent systems under unknown discrete-time dynamics and limited sensing conditions. The paper introduces discrete-time extensions of control barrier functions and leverages a policy optimization technique to jointly learn safe and performant policies without requiring a predefined nominal policy. DGPPO is evaluated across multiople simulation environments, demonstrating superior safety and cost performance compared to baselines while being robust to hyperparameter changes.

Comments: We received 3 expert reviews, with the scores 6, 6, 8, and the average score is 6.67. The reviewers are generally positive about the algorithmic and technical contributions of this paper. The discrete graph formulation and their integration with reinforcement learning overcomes many limitations of continuous-time CBFs. The technical analysis is rigorous, with results that ensure forward invariance of safety constraints and address the challenges of unknown dynamics in MAS.  Simulations demonstrate near-perfect safety rates as a validation of the proposed approach.  The paper is also well-written.

The reviewers have also provided multiple comments to strengthen the paper. One main comment was to include additional experiments that demonstrate the scalability. Another comment is to further validate the claim that the proposed method is robust to hyperparameters changes. I am glad to see that the authors have already addressed these comments during the rebuttal and the reviewers have acknowledged their efforts.  I recommend the authors update the paper by accommodating these suggestions.

**Additional Comments On Reviewer Discussion:**

Please see the "Comments" in the meta-review.

---

### Decision · Program_Chairs · 2025-01-22

Accept (Poster)